# Protein composition of axonal dopamine release sites in the striatum

Lauren Kershberg, Aditi Banerjee, Pascal S Kaeser*

Department of Neurobiology, Harvard Medical School, Boston, United States

**Abstract** Dopamine is an important modulator of cognition and movement. We recently found that evoked dopamine secretion is fast and relies on active zone-like release sites. Here, we used in vivo biotin identification (iBioID) proximity proteomics in mouse striatum to assess which proteins are present at these sites. Using three release site baits, we identified proteins that are enriched over the general dopamine axonal protein content, and they fell into several categories, including active zone, $Ca^{2+}$ regulatory, and synaptic vesicle proteins. We also detected many proteins not previously associated with vesicular exocytosis. Knockout of the presynaptic organizer protein RIM strongly decreased the hit number obtained with iBioID, while Synaptotagmin-1 knockout did not. α-Synuclein, a protein linked to Parkinson's disease, was enriched at release sites, and its enrichment was lost in both tested mutants. We conclude that RIM organizes scaffolded dopamine release sites and provide a proteomic assessment of the composition of these sites.

## Editor's evaluation

Using a smart proximity labeling approach, the protein composition of dopaminergic neurotransmitter release sites was determined in striatal axons. Using mice in which release sites were disrupted as control, the authors identified not only established components of the secretory machinery but also many new proteins whose function awaits further characterization. The datasets provided are of very high quality and provide an important foundation for studies on the dopaminergic exocytotic machinery.

**\*For correspondence:**
kaeser@hms.harvard.edu

**Competing interest:** The authors declare that no competing interests exist.

## Introduction

Dopamine is a critical neuromodulator and regulates target cells through G-protein-coupled receptor (GPCR) signaling (*Berke, 2018*; *Liu et al., 2021a*; *Surmeier et al., 2014*). In the vertebrate brain, midbrain dopamine neurons form a crucial regulatory system. The dopamine neuron somata are located in the substantia nigra pars compacta (SNc) and the ventral tegmental area (VTA), and one prominent target area of their axons is the striatum. In the striatum, dopamine is thought to act as a volume transmitter because only a small percentage of dopamine varicosities are directly apposed to postsynaptic specializations (*Descarries et al., 1996*; *Wildenberg et al., 2021*) and because dopamine receptors are found mostly extrasynaptically (*Liu et al., 2021a*; *Rice et al., 2011*; *Sesack et al., 1994*; *Uchigashima et al., 2016*; *Yung et al., 1995*). Hence, dopamine likely signals by diffusing through the extracellular striatal space before initiating responses in target cells. While much is understood about the mechanisms and molecules involved in the synaptic release of classical neurotransmitters, the machinery mediating the release of dopamine is less well known.

Spatial and temporal precision of vesicular exocytosis at classical synapses is established by the active zone, a conserved molecular machine that docks synaptic vesicles at the presynaptic plasma membrane close to voltage-gated $Ca^{2+}$ channels, primes the vesicles for release, and aligns these processes with postsynaptic receptor clusters (*Biederer et al., 2017*; *Südhof, 2012*). Several lines of evidence point to the existence of active zone-like protein complexes for the control of striatal

dopamine release. First, dopamine release is rapid, has a high release probability, and occurs at small hotspots (*Banerjee et al., 2022*; *Banerjee et al., 2020*; *Beyene et al., 2019*; *Liu et al., 2018*; *Marcott et al., 2014*; *Patriarchi et al., 2018*; *Silm et al., 2019*; *Wang et al., 2014*; *Zych and Ford, 2022*), indicating the need for vesicle tethering close to $Ca^{2+}$ channels before the stimulus arrives. Second, at least some dopamine receptors respond rapidly to local, high concentrations of dopamine (*Beckstead et al., 2004*; *Condon et al., 2021*; *Courtney and Ford, 2014*; *Gantz et al., 2013*; *Marcott et al., 2014*), which likely necessitates synchrony of vesicular release. Third, dopamine neuron activity and secretion often correlate with behavior on fast time scales (*Bova et al., 2020*; *Chaudhury et al., 2013*; *da Silva et al., 2018*; *Hamilos et al., 2021*; *Hollerman and Schultz, 1998*; *Howe and Dombeck, 2016*; *Jin and Costa, 2010*; *Schultz et al., 1997*), suggesting the need for precise signaling mechanisms. Taken together these properties indicate the presence of protein machinery for rapid and efficient release of dopamine.

Indeed, our recent work has found that proteins for the control of spatial and temporal precision of synaptic vesicle exocytosis are important for evoked dopamine release. These proteins include the active zone scaffolds RIM and Liprin-α, the priming protein Munc13, and the fast $Ca^{2+}$ sensor Synaptotagmin-1 (Syt-1) (*Banerjee et al., 2022*; *Banerjee et al., 2020*; *Lebowitz et al., 2022*; *Liu et al., 2018*; *Robinson et al., 2019*). Conversely, the active zone scaffolds RIM-BP and ELKS are dispensable for axonal dopamine release, even though at least ELKS is present at these sites (*Banerjee et al., 2022*; *Liu et al., 2018*). Striatal dopamine release only partially depends on voltage-gated $Ca^{2+}$ channels of the $Ca_V2$ family, channels that are required for stimulated release at most synapses (*Brimblecombe et al., 2015*; *Held et al., 2020*; *Liu et al., 2022*; *Luebke et al., 1993*; *Takahashi and Momiyama, 1993*). Thus, there appear to be both similarities and differences in the secretory machines in dopamine varicosities compared with synapses. However, only a handful of proteins important for secretion are known to mediate dopamine release, and many pieces of the underlying secretory machine remain unidentified.

We here characterized the composition of dopamine release sites using an unbiased proteomic approach. We adapted in vivo biotin identification (iBioID), a method that allows for purifying proteins in close proximity to a marker protein (*Kim et al., 2014*; *Roux et al., 2012*; *Uezu et al., 2016*). Using three different release site bait proteins, we found that 527 proteins were present at these sites when we applied a threshold of ≥2.0-fold enrichment over soluble axonal proteins. A total of 190 proteins were enriched in multiple bait conditions and 41 proteins with all three baits. Many proteins with previously known functions at classical synapses were identified, including active zone, $Ca^{2+}$ regulatory, and synaptic vesicle proteins, as well as proteins that presently do not have defined roles in axonal neurotransmitter secretion. We also tested whether structural scaffolding and dopamine release are important for the composition of these secretory sites. Conditional, dopamine neuron-specific knockout of the presynaptic organizer protein RIM strongly decreased the number of detected proteins, while that of the $Ca^{2+}$ sensor Syt-1 to abolish synchronous dopamine release did not. We conclude that dopamine release sites are organized structures controlled by the scaffolding protein RIM, and we provide a proteomic assessment of the content of these release sites.

## Results

### Proximity proteomics to assess dopamine release site composition in the mouse striatum

With the overall goal to generate a comprehensive proteome of release sites in striatal dopamine axons, we adapted a proximity proteomic approach. We fused a promiscuous version of the biotinylase BirA (also called BirA*) (*Roux et al., 2012*; *Uezu et al., 2016*) to several proteins associated with dopamine release sites and expressed these bait proteins Cre-dependently using AAVs specifically in dopamine neurons of DAT[IRES-Cre] mice (*Bäckman et al., 2006*). To locally biotinylate proteins in vivo, we provided excess biotin through subcutaneous injection for 7 days. We then performed affinity purification of the biotinylated proteins from striatal homogenates and identified them by mass spectrometry (*Figure 1A*). The striatal dopamine axons are particularly well suited for this approach because they are elaborately branched in the striatum and can be easily separated from their midbrain somata and dendrites during tissue dissection, limiting confounds arising from the co-purification of somatodendritic proteins.

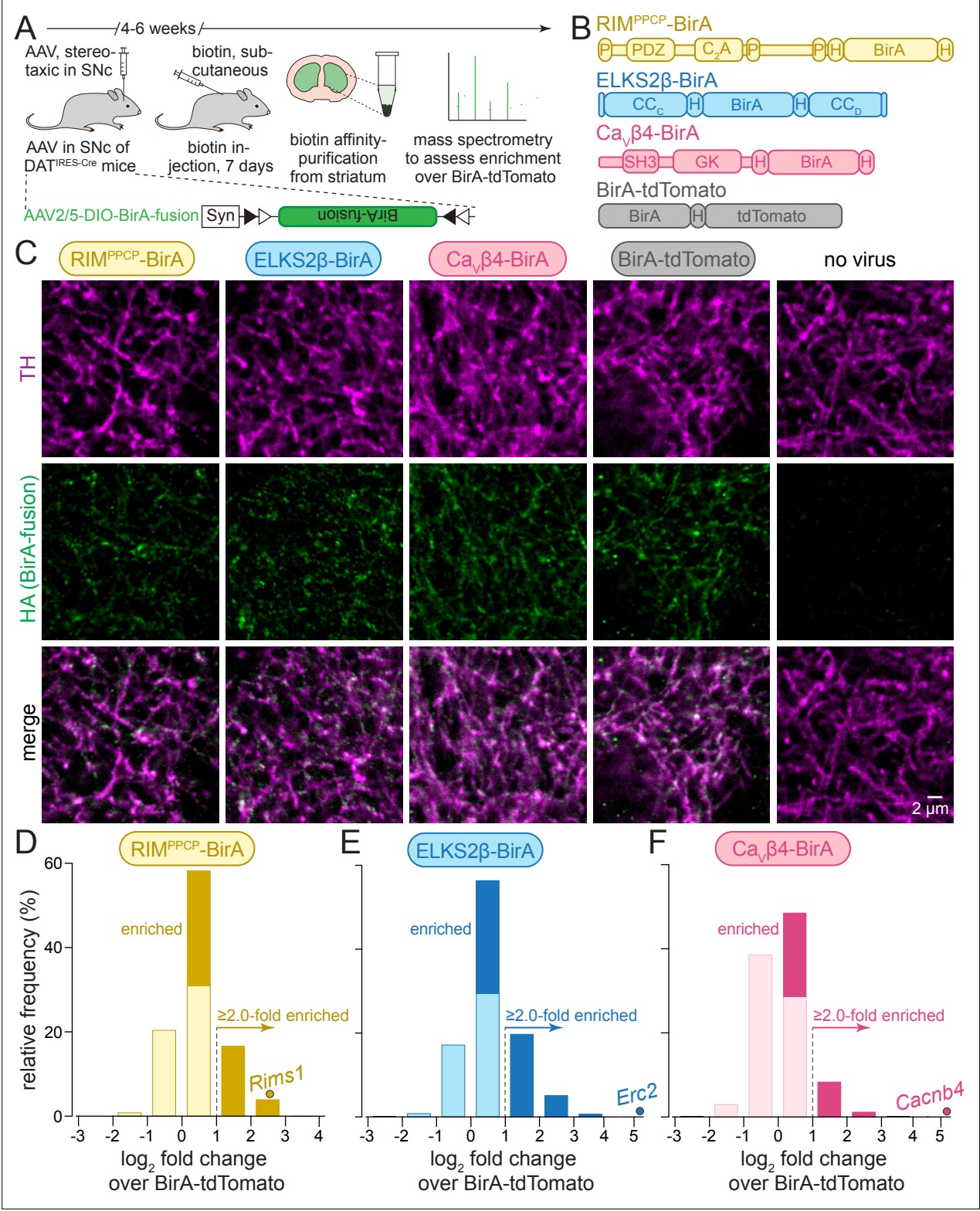

**Figure 1.** In vivo biotin identification (iBioID) for release site proteins in dopamine axons of the mouse striatum. (**A**) Schematic of the experiment with AAVs for Cre-dependent BirA fusion protein expression (AAV2/5-DIO-BirA-fusion), followed by in vivo biotinylation, affinity purification, and analyses by mass spectrometry. (**B**) Overview of BirA fusion proteins expressed Cre-dependently with AAVs in midbrain dopamine neurons of DAT[IRES-Cre] mice, P: proline-rich motif, H: hemagglutinin (HA) tag. Each mouse expressed one of the three BirA baits (RIM[PPCP]-BirA, ELKS2β-BirA, and Ca$_V$β4-BirA) or BirA-tdTomato (to generate a proteome for normalization). (**C**) Sample confocal images of striatal slices of DAT[IRES-Cre] mice expressing the BirA fusion proteins

*Figure 1 continued*

shown in (**B**), slices were stained with anti-TH antibodies and anti-HA antibodies to label dopamine axons and BirA fusion proteins, respectively. Images are from an experiment in which all constructs were imaged in the same session and with the same settings, and adjusted identically for display, except for ELKS2β-BirA. Images for ELKS2β-BirA were acquired in a separate experiment and with its own control; these images were adjusted for brightness and contrast slightly differently to match overall appearance, and adjustments were identical to those made to its own control acquired at the same time and shown in *Figure 1—figure supplement 1*. A sample image area is shown from 3 to 5 overview images per mouse and condition, each experiment was repeated in ≥3 mice. (**D–F**) Protein enrichment in BirA bait conditions over BirA-tdTomato. $Log_2$ fold change values are plotted as frequency histograms. Values at or below 0 represent proteins that are equal to or lower than in the BirA-tdTomato condition (light colors), values >0 represent proteins that are higher than in the BirA-tdTomato condition (saturated colors), and values ≥1 represent hits with ≥2.0-fold enrichment; (**D**) 1306 total proteins identified, 659 proteins with $log_2$ fold change >0, and 269 proteins with $log_2$ fold change ≥1 (hits); (**E**) 1496/805/382; (**F**) 1168/354/114. The gene encoding the protein that was used as a bait is shown as a dot and labeled individually in each panel; (**D**) 4 independent repeats (12 striata each); (**E**) 4 (12/12/12/10); (**F**) 4 (12/12/12/10). For sample images of the negative control for ELKS2β-BirA, see *Figure 1—figure supplement 1*; for pilot iBioID experiments and assessment of self-biotinylation, see *Figure 1—figure supplement 2*; for a table of all proteins identified, see *Source data 1A*; for cDNA sequences used to generate the fusion proteins, see *Source data 1B*.

The online version of this article includes the following source data and figure supplement(s) for figure 1:

**Figure supplement 1.** Sample images for ELKS2β-BirA expression.

**Figure supplement 2.** BirA fusion proteins are self-biotinylated and purified with iBioID.

**Figure supplement 2—source data 1.** Western blots for *Figure 1 – figure supplement 2A*.

**Figure supplement 2—source data 2.** Western blots for *Figure 1 – figure supplement 2B*.

We first generated a series of AAVs to express bait proteins in a Cre-dependent manner. In these bait proteins, BirA was fused to RIM or ELKS, active zone proteins associated with dopamine release sites (*Banerjee et al., 2022*; *Liu et al., 2018*), or to the β4 subunit of voltage-gated $Ca^{2+}$ channels ($Ca_V$β4), a component of $Ca_V$ $Ca^{2+}$ channel complexes that are present at release sites (*Brimblecombe et al., 2015*; *Dolphin, 2003*; *Held et al., 2020*; *Tan et al., 2022b*). Full-length active zone proteins fused to BirA exceed the packaging limit of AAVs. Considering this limitation, the following constructs were selected from initial trial experiments (*Figure 1B*): RIM^PPCP-BirA, containing the central region of RIM (PDZ and $C_2$A domains with endogenous linker sequences and proline-rich motifs) that is important for its active zone localization, with BirA inserted at the C-terminus (*Kaeser et al., 2011*; *Tan et al., 2022a*; *Tang et al., 2016*; *Wu et al., 2019*); ELKS2β-BirA, a short version of ELKS with BirA inserted between the $CC_C$ and $CC_D$ domains (*Held et al., 2016*; *Kaeser et al., 2009*; *Liu et al., 2014*); and $Ca_V$β4-BirA, a $Ca_V$ subunit with BirA inserted at the C-terminus (*Dolphin, 2003*; *Held et al., 2020*; *Tan et al., 2022b*). In all experiments, we used BirA fused to tdTomato (BirA-tdTomato) that localizes throughout the axon for calculating enrichment. All constructs also contained hemagglutinin (HA)-tags for identification with HA antibodies.

To assess the axonal presence of these BirA fusion proteins, we injected the AAVs into the midbrain of DAT^IRES-Cre mice. After 6 weeks, we perfused the mice and stained striatal brain slices with antibodies against HA (to label the fusion proteins) and antibodies against tyrosine hydroxylase (TH, to label dopamine axons) (*Figure 1C*, *Figure 1—figure supplement 1*). The fusion proteins were detected within TH-labeled axons. RIM^PPCP-BirA and ELKS2β-BirA were present in small punctate structures consistent with release site localization (*Liu et al., 2018*). $Ca_V$β4 appeared somewhat more widespread, similar to $Ca^{2+}$ entry in these axons (*Pereira et al., 2016*), and BirA-tdTomato was broadly overlapping with TH, likely reflecting a soluble axonal localization. We previously performed superresolution microscopy to assess the localization of select endogenous dopamine axonal proteins (*Banerjee et al., 2022*; *Banerjee et al., 2020*; *Liu et al., 2018*). High-quality super-resolved images could not be obtained here, likely because of the specific antibodies and conditions that were needed. In vivo biotinylation was next confirmed for the BirA fusion proteins with or without biotin injections (for 7 days) followed by pilot biotin-affinity purifications and Western blotting of the purified fractions (*Figure 1—figure supplement 2* and 'Materials and methods'). This established that biotinylation and protein purification are efficient, and that background biotinylation in the absence of biotin is limited.

To determine dopamine release site composition, we performed iBioID using these three BirA bait proteins and BirA-tdTomato expressed in DAT^IRES-Cre mice (*Figure 1A*). After 4–6 weeks of expression, biotin injections were done on seven consecutive days, and 10–12 striata per condition and repeat were dissected and homogenized. Biotinylated proteins were then isolated using affinity purification. The eluates were depleted with antibodies against pyruvate carboxylase (PC), an endogenously

biotinylated protein, before assessment of protein content with mass spectrometry. In total, four independent biological repeats per BirA fusion protein were performed (i.e., four times 10–12 striata per condition, conditions: RIM$^{PPCP}$-BirA, ELKS2β-BirA, Ca$_V$β4-BirA, BirA-tdTomato, also see 'Materials and methods'). The four repeats were run as two large experiments for in vivo biotinylation, biotin affinity purification, and mass spectrometry; two repeats were run in parallel in each experiment, and the two experiments were run approximately 1 year apart from one another (the second experiment also contained analyses of conditional knockout mice described below).

Peptides from 1969 proteins were identified in at least one of the BirA conditions. A total of 300 (15%) of those proteins are mitochondrial proteins, which is common in iBioID, and these proteins were removed from further analyses unless noted otherwise, as has been done before (*Loh et al., 2016*; *Uezu et al., 2016*). To identify proteins enriched in the immediate vicinity of the bait proteins (RIM$^{PPCP}$-BirA, ELKS2β-BirA, and Ca$_V$β4-BirA), the number of peptides identified for each protein and bait was normalized to the number of peptides found for the same protein in the BirA-tdTomato condition, effectively generating a ratio of enrichment over soluble dopamine axonal protein content as fold change. For proteins that were not detected with all BirA fusions, we assigned an average value of 0.5 peptides so that we could calculate fold change values for proteins that otherwise had a 0 in the denominator and log$_2$ of fold change for all detected proteins. After producing log$_2$ of the fold change, values >0 represent proteins detected at levels higher than in the BirA-tdTomato condition (*Figure 1D–F*, saturated colors), while proteins <0 (light colors) were below. For the main analyses, 'hits' were proteins enriched ≥2.0-fold and have log$_2$ fold change values of ≥1 (*Figure 1D–F*). These proteins are included in the analyses presented in *Figure 2*. Analyses with alternate enrichment thresholds are shown in *Figure 2—figure supplements 1 and 2*. These applied cutoffs are similar to previous studies using iBioID (*Takano et al., 2020*; *Uezu et al., 2016*).

For RIM$^{PPCP}$-BirA (*Figure 1D*) and ELKS2β-BirA (*Figure 1E*), 50% and 54% of the detected proteins were higher than with BirA-tdTomato, and 20% and 25% were above the 2.0-fold enrichment threshold, respectively. Only 30% of all proteins detected in the Ca$_V$β4-BirA condition were higher compared to BirA-tdTomato, and 9% were enriched ≥2.0-fold. These observations generally align with the morphological data (*Figure 1C*), where Ca$_V$β4-BirA appeared more widely expressed than RIM$^{PPCP}$-BirA and ELKS2β-BirA. They are also consistent with functional analyses that revealed that Ca$^{2+}$ entry in dopamine axons is widespread (*Pereira et al., 2016*), while active zone proteins (*Banerjee et al., 2022*; *Liu et al., 2018*) and release (*Pereira et al., 2016*) are only detected in 20–30% of the varicosities. In each condition, the protein that was used as a BirA bait was robustly enriched (individually labeled proteins in *Figure 1D–F*). This further establishes the approach, as self-biotinylation of the BirA fusion proteins should be high compared to other proteins if the labeling radius is small, which has been estimated to be 10–50 nm (*Kim et al., 2014*; *Roux et al., 2012*).

## Assessment of the protein composition of release sites in dopamine axons

We next constructed Venn diagrams with proteins that were above the 2.0-fold enrichment threshold to assess the overlap of hits between the different bait conditions. In total, there were 527 proteins ≥2.0-fold enriched (*Figure 2A*). The enrichment compared to BirA-tdTomato indicates that these proteins have a localization that is restricted compared to general dopamine axonal proteins. Of the 527 proteins, 190 were enriched in more than one condition and 41 were enriched with all three baits, RIM$^{PPCP}$-BirA, ELKS2β-BirA, and Ca$_V$β4-BirA.

Given that the BirA baits are proteins with roles in transmitter secretion and are expressed in dopamine axons, it is expected that proteins with known roles in neurotransmitter release are enriched in the iBioID dataset. We assessed the hits by evaluating their cellular compartment annotations in SynGO, an expert-curated database that assigns localization and function of synaptic genes (*Koopmans et al., 2019*). Of the 527 enriched proteins, 194 (37%) have one or multiple synaptic cellular compartment annotations in SynGO (*Figure 2B*), reflecting that their synaptic localization has been established. SynGO does not distinguish between synapse or neurotransmitter type, but instead broadly determines whether there is evidence to support the presence or function of a given protein at synapses in general (*Koopmans et al., 2019*).

To assess how the enrichment threshold of ≥2.0 influenced this overall assessment of the release site proteome, we constructed Venn diagrams with alternate enrichment thresholds of ≥1.5

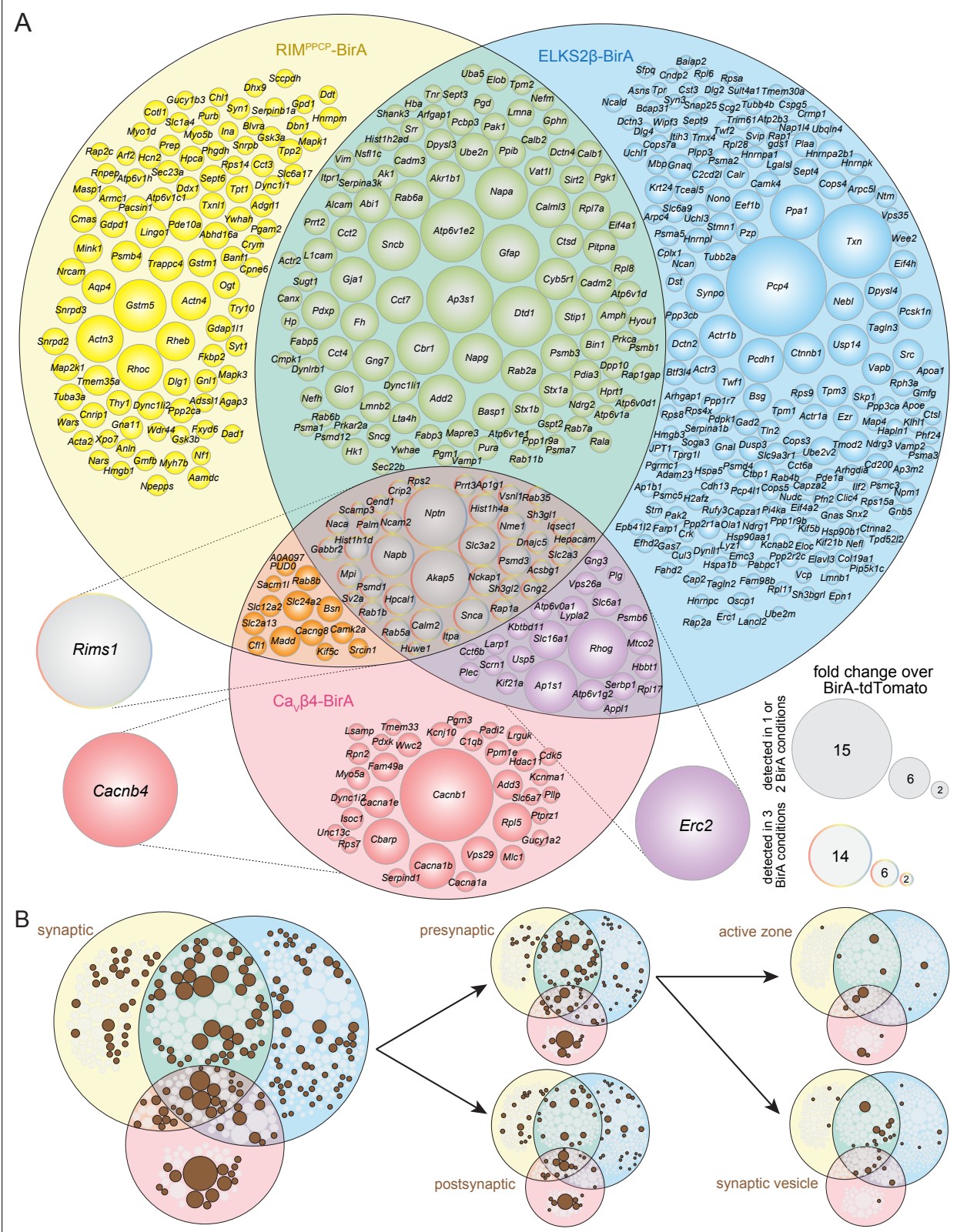

**Figure 2.** Protein composition of release sites in dopamine axons. (**A**) Venn diagram listing genes that encode protein hits enriched at least 2.0-fold over BirA-tdTomato with RIM$^{PPCP}$-BirA, ELKS2β-BirA, or Ca$_V$β4-BirA iBioID baits. Bait-encoding genes are shown outside the Venn diagram in the color of the corresponding part of the diagram. Enrichment is shown as the average of 4 independent repeats run in 2 mass spectrometry sessions (repeats 1 and 2 in the first session, repeats 3 and 4 in the second) and was calculated as described in the 'Materials and methods.' Circle size reflects average

*Figure 2 continued on next page*

*Figure 2 continued*

enrichment across repeats (key on the bottom right). For proteins enriched in multiple bait conditions, circle size corresponds to the bait condition with the largest enrichment. (**B**) Hits from (**A**) that have cellular compartment annotations in the SynGO database (*Koopmans et al., 2019*) are colored in brown and are classified into increasingly specific SynGO subcategories, subcategorizations are not mutually exclusive; proteins used as baits are not included. For a list of hits, see *Source data 1C*; for Venn diagrams constructed with 1.5- and 2.5-fold enrichment thresholds, see *Figure 2—figure supplements 1 and 2*; for a comparison of SynGO cellular compartment annotations across thresholds, see *Figure 2—figure supplement 3*; and for assessment of Neuroplastin (*Nptn*) localization in striatal synaptosomes, see *Figure 2—figure supplement 4*.

The online version of this article includes the following figure supplement(s) for figure 2:

**Figure supplement 1.** Venn diagram of the dopamine release site proteome with a 1.5-fold enrichment threshold.

**Figure supplement 2.** Venn diagram of the dopamine release site proteome with a 2.5-fold enrichment threshold.

**Figure supplement 3.** Assessment of release site hits listed in SynGO across enrichment thresholds.

**Figure supplement 4.** Neuroplastin antibody labeling is enhanced in Bassoon-containing dopaminergic synaptosomes.

(*Figure 2—figure supplement 1*) or ≥2.5 (*Figure 2—figure supplement 2*). Naturally, the overall number of proteins that reached threshold varied, with 991 proteins being ≥1.5-fold and 348 proteins ≥2.5-fold enriched, compared to the 527 proteins detected with the ≥2.0-fold enrichment threshold. Changing enrichment thresholds did not increase the percentage of synaptic proteins as identified via SynGO (31% for ≥1.5 and 33% for ≥2.5 thresholds, respectively, *Figure 2—figure supplement 3*), justifying the use of ≥2.0 as the threshold for further analyses.

We next categorized the proteins based on whether they are known to be localized pre- or post-synaptically in order to assess which types of synaptic proteins were enriched. Proteins with SynGO annotations may have multiple annotations and can be both pre- and postsynaptic, but do not require a more specific annotation. A total of 102 proteins (19% of the entire dataset, 53% of the SynGO synaptic proteins) have a presynaptic annotation with many having additional specific assignments to the active zone or synaptic vesicles, while 87 (17% of the entire dataset, 45% of the SynGO synaptic proteins) have postsynaptic annotations (*Figure 2B*). Also, 43 proteins (8% of the entire dataset, 22% of the SynGO synaptic proteins) are annotated both pre- and postsynaptically.

If the iBioID approach used here enriched proteins at sites of dopamine release, it would be expected that proteins previously shown to localize to dopamine release sites are enriched. Indeed, RIM1 (*Rims1*), ELKS2 (*Erc2*), and Bassoon (*Bsn*), proteins that can be used as markers for dopamine release sites (*Banerjee et al., 2022*; *Liu et al., 2018*) and that were shown to be associated with synaptic vesicle docking sites using a different proteomic approach (*Boyken et al., 2013*), were all enriched across multiple conditions (*Figure 2A*).

Proteomic assessment of release site composition, like the one described here, can serve to identify new putative components. Of the 527 enriched proteins with the ≥2.0-fold threshold, 333 (63%) do not have a SynGO annotation (*Figure 2B*), and 46 proteins are annotated broadly as 'synaptic' without more exact specification. Additionally, for the characterization of the dopamine axon secretory machinery, proteins known to be associated with classical synaptic release that have not been implicated in dopamine release can also be identified. One example of the latter category is Neuroplastin (*Nptn*), a transmembrane protein with pre- and postsynaptic roles at synapses (*Beesley et al., 2014*; *Boyken et al., 2013*; *Schmidt et al., 2017*; *Smalla et al., 2000*). Neuroplastin is strongly enriched across conditions (*Figure 2A*), but its association with dopamine release sites has – to our knowledge – not been described. To assess Neuroplastin localization with a secondary approach, we prepared striatal synaptosomes as we described before (*Banerjee et al., 2022*; *Banerjee et al., 2020*; *Liu et al., 2018*) and stained them with antibodies against TH to mark dopamine synaptosomes, Bassoon to label release sites, and Neuroplastin. In particles labeled with TH, Neuroplastin antibody staining intensity was significantly higher when Bassoon was present (*Figure 2—figure supplement 4*). These results suggest that Neuroplastin is enriched in proximity to release sites of dopamine axons and provide an independent approach that helps validate iBioID for this specific hit.

## Ablation of RIM, but not of Synaptotagmin-1, strongly decreased the number of proteins detected in the dopamine release site proteome

We next asked whether removing proteins important for dopamine release affects the composition of the dopamine release site proteome. Previous work has established genetic strategies to abolish

(1) evoked dopamine release by removing the presynaptic scaffolding protein RIM (Banerjee et al., 2022; *Liu et al., 2018*; *Robinson et al., 2019*) or (2) synchronous dopamine release by removing the fast $Ca^{2+}$ sensor Syt-1 (*Banerjee et al., 2020*; *Lebowitz et al., 2022*). We crossed mice with floxed alleles for RIM1 and RIM2 or for Syt-1 to DAT^IRES-Cre^ mice (*Bäckman et al., 2006*; *Kaeser et al., 2011*; *Banerjee et al., 2020*; *Kochubey et al., 2016*; *Liu et al., 2018*; *Zhou et al., 2015*) to remove these proteins from dopamine neurons (*Figure 3A*, RIM cKO^DA^ or Syt-1 cKO^DA^, respectively). We then performed iBioID analogous to the proteome described in *Figures 1 and 2* (called 'control proteome' from this point forward), with the modification that in each mutant we completed two independent repeats instead of four (i.e., two times 10–12 striata for each of the bait conditions in RIM cKO^DA^ and Syt-1 cKO^DA^ mice) due to the genetic complexity and volume of the experiment.

We first compared the BirA-tdTomato condition in control, RIM cKO^DA^, and Syt-1 cKO^DA^ and found that both the number of identified proteins and the peptide counts were similar across experiments (*Figure 3B*). Hence, removal of RIM or Syt-1 from dopamine neurons did not cause strong disruptions in the overall dopamine axon protein content in the striatum. The presence of similar amounts of axonal material is consistent with previous work that found TH axon density to be similar to control mice in these mutants (*Banerjee et al., 2020*; *Liu et al., 2018*).

We next assessed fold change over tdTomato for each of the BirA baits and each mutant. In RIM cKO^DA^ mice, 31% (RIM^PPCP^-BirA), 12% (ELKS2β-BirA), and 13% (Ca$_V$β4-BirA) of the identified proteins were higher than with BirA-tdTomato, and 14% (RIM^PPCP^-BirA), 6% (ELKS2β-BirA), and 4% (Ca$_V$β4-BirA) reached the ≥2.0-fold enrichment threshold to be considered hits (*Figure 3C–E*). These percentages are overall lower than in the control proteome (*Figure 1D–F*). The protein enrichment over BirA-tdTomato in Syt-1 cKO^DA^ mice was more similar to the control proteome (*Figure 3F–H*, RIM^PPCP^-BirA: 42% higher than with BirA-tdTomato and 20% ≥2.0-fold enriched; ELKS2β-BirA: 56% and 30%, Ca$_V$β4-BirA 53% and 24%). We conclude that in RIM cKO^DA^ mice, release site protein enrichment close to the baits is disrupted compared to control or Syt-1 cKO^DA^ mice. This suggests that RIM removal disrupts mechanisms important for release site scaffolding. Abolishing synchronous dopamine release by Syt-1 cKO^DA^ has no strong effects on the overall number of proteins detected at release sites in dopamine axons with iBioID.

We next assessed hits in more detail by generating and analyzing Venn diagrams for each mutant. The RIM cKO^DA^ dataset contained a total of 198 hits (*Figure 3—figure supplement 1*), compared to the 527 hits in the control proteome (*Figure 2*). RIM1 enrichment was absent from all conditions except for RIM^PPCP^-BirA, which likely reflects self-biotinylation of the bait. It is noteworthy that 49% of all hits (98 out of 198) in the RIM cKO^DA^ dataset came from the RIM^PPCP^-BirA bait only (*Figure 3—figure supplement 1*, yellow circle), indicating that expression of RIM^PPCP^, the central region of RIM containing its scaffolding domains (*Figure 1B*), may restore some scaffolding deficits caused by RIM cKO^DA^. In Syt-1 cKO^DA^ mice, 450 hits were detected with the 2.0-fold enrichment threshold (*Figure 3—figure supplement 2*), similar to the 527 hits in the control dataset, and 104 (23%) of the hits were at least 2.0-fold enriched for all three BirA baits, including RIM (*Figure 3—figure supplement 2*, center). Changing enrichment thresholds to ≥1.5-fold (RIM cKO^DA^: 268 hits; Syt-1 cKO^DA^ 602 hits) or ≥2.5-fold (RIM cKO^DA^: 82 hits; Syt-1 cKO^DA^ 252 hits) substantiated the conclusion that RIM but not Syt-1 ablation strongly decreased the overall hit number.

To characterize which proteins were depleted from the release site proteome in the RIM cKO^DA^ mice, we first used SynGO to categorize all identified proteins in each dataset into synaptic proteins (*Figure 4A*, gray bars), presynaptic proteins (*Figure 4B*), and active zone proteins (*Figure 4C*). We then plotted the $\log_2$ of the fold change of the average of all annotated proteins in each category. In the control dataset (gray bars), synaptic proteins were enriched as expected (*Figure 4A*), and the extent of enrichment had a tendency to increase as the SynGO annotation became more specific (to presynaptic, *Figure 4B*, and to active zone, *Figure 4C*). These effects not only disappeared in RIM cKO^DA^ mice, but instead reverted (*Figure 4A–C*, green bars), and for ELKS2β-BirA and Ca$_V$β4-BirA, proteins in all three SynGO categories appeared depleted ($\log_2$ fold change values <0). The depletion is absent in the RIM^PPCP^-BirA condition in RIM cKO^DA^ mice, supporting that release site scaffolding is partially restored. In contrast to RIM cKO^DA^, the proteins found in the Syt-1 cKO^DA^ dataset show an enrichment that is overall relatively similar to the control dataset with all conditions having $\log_2$ fold change values >0 (*Figure 4A–C*, maroon bars), supporting that abolishing

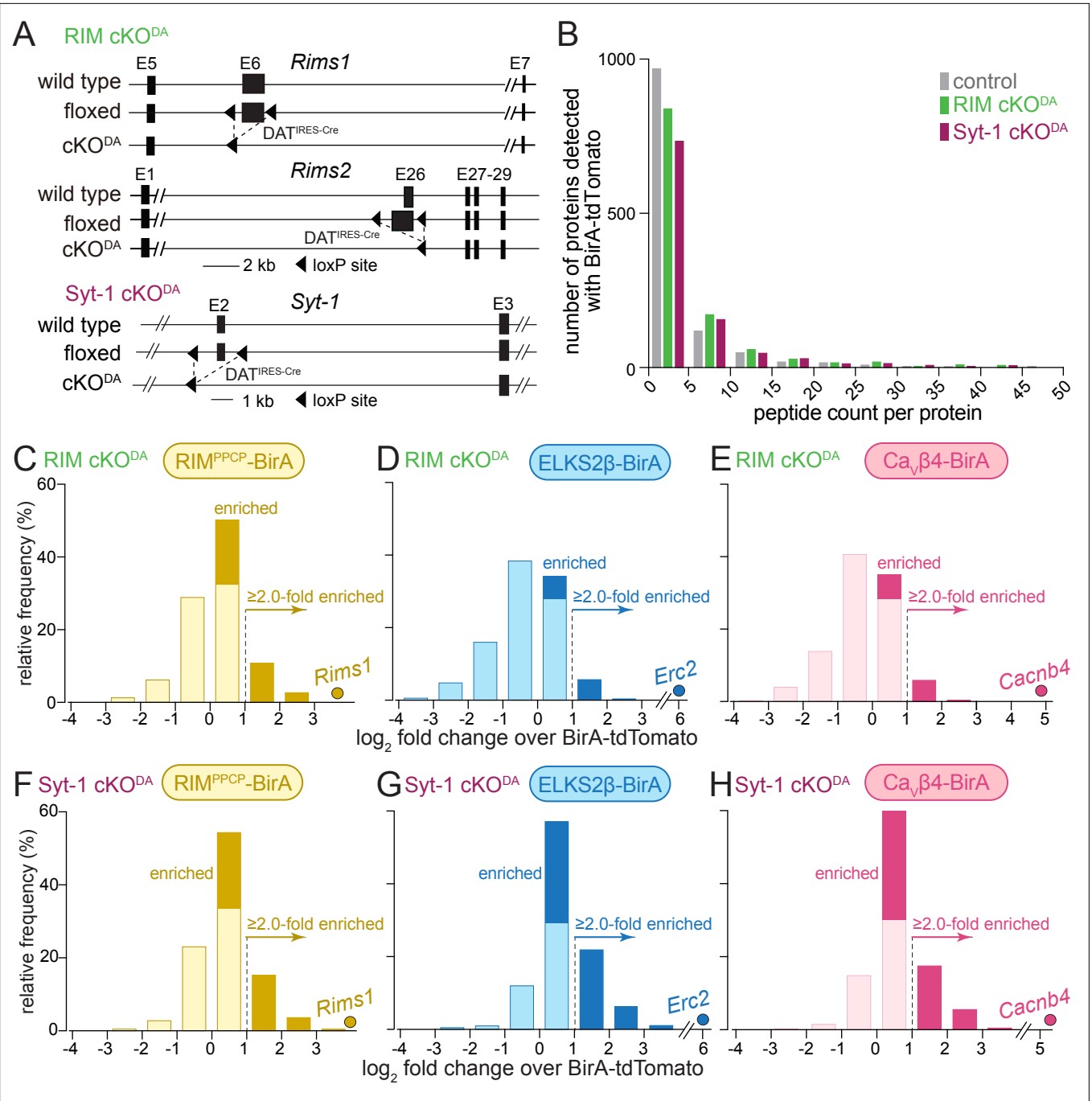

**Figure 3.** Enrichment of release site proteins after conditional ablation of RIM or of Synaptotagmin-1 from dopamine neurons. (**A**) Strategy for ablation of RIM1 and RIM2 (RIM cKO$^{DA}$) or Synaptotagmin-1 (Syt-1 cKO$^{DA}$) from dopamine neurons using conditional mouse genetics (*Banerjee et al., 2020*; *Liu et al., 2018*). (**B**) The average number of peptides per protein in bins of five detected with BirA-tdTomato. The control is from data in *Figures 1 and 2* and is the average peptide count of 4 repeats, the proteomes from RIM cKO$^{DA}$ and Syt-1 cKO$^{DA}$ are an average peptide count of 2 repeats each. The x-axis is cut at a peptide count of 50 covering >99% of the detected proteins. Average number of detected proteins: control, 1195; RIM cKO$^{DA}$, 1155; Syt-1 cKO$^{DA}$, 1011. (**C–E**) Protein enrichment in BirA bait conditions over BirA-tdTomato in RIM cKO$^{DA}$ mice. Log$_2$ fold change values are plotted as frequency histograms. Values at or below 0 represent proteins that are equal to or lower than in the BirA-tdTomato condition (light colors), values >0 represent proteins that are higher than in the BirA-tdTomato condition (saturated colors), and values ≥1 represent hits with ≥2.0-fold enrichment; (**C**) 1089 total proteins identified, 334 proteins with log$_2$ fold change >0, and 149 proteins with log$_2$ fold change ≥1 (hits); (**D**) 1003/123/62; (**E**) 1008/130/65. The gene encoding the protein that was used as a bait is shown as a dot and labeled individually in each panel: (**C**) 2 independent repeats (10/12 striata); (**D**) 2 (12/10); (**E**) 2 (10/12). (**F–H**) Same as (**C–E**), but for Syt-1 cKO$^{DA}$ mice; (**F**) 1017 total proteins identified, 424 proteins with log$_2$ fold change >0, and 199 proteins with log$_2$ fold change ≥1 (hits); (**G**) 1103/619/327; (**H**) 1016/544/246, (**F**) 2 independent repeats (10/12 striata); (**G**) 2 (12/12); (**H**) 2 (12/12).

*Figure 3 continued on next page*

*Figure 3 continued*

Two-way ANOVA was used in (**B**) (genotype: p>0.6, peptide count: p<0.001). For a table of all peptides identified in RIM cKO[DA], see **Source data 1F**; for a table of all peptides identified in Syt-1 cKO[DA], see **Source data 1G**; and for Venn diagrams of release site protein enrichment, see **Figure 3—figure supplement 1** for RIM cKO[DA] and **Figure 3—figure supplement 2** for Syt-1 cKO[DA].

The online version of this article includes the following figure supplement(s) for figure 3:

**Figure supplement 1.** Venn diagrams for the RIM cKO[DA] dataset.

**Figure supplement 2.** Venn diagrams for the Syt-1 cKO[DA] dataset.

synchronous dopamine release does not strongly disrupt overall release site protein content. These effects were overall similar for proteins annotated as synaptic vesicle or postsynaptic proteins in SynGO (*Figure 4—figure supplement 1*).

We next mapped the ≥2.0-fold enriched proteins detected in each mutant onto the Venn diagram of the control proteome shown in *Figure 2*. Only 15% of proteins in the control dataset were also enriched in the RIM cKO[DA] dataset (*Figure 4D*), and 49% of those proteins stem from the RIM[PPCP]-BirA condition (*Figure 4D*, circles with yellow outline). In contrast, 37% of the proteins in the control dataset were also enriched in the Syt-1 cKO[DA] dataset (*Figure 4E*). The observation that there is no full overlap between the control and Syt-1 cKO[DA] datasets is likely due to a combination of factors. Examples are that the approach may not be saturating and may not detect all release site proteins, that the mutant data were derived from fewer repeats, and that Syt-1 cKO[DA] may influence the exact composition but not the overall extent of the release site proteome.

Finally, to assess the organization of the proteins and their interactions in the various release site proteomes, we used the STRING database (*Snel et al., 2000*; *Szklarczyk et al., 2019*). STRING combines both theoretical predictions and empirical data to generate maps of protein–protein functional and physical interactions. We selected the proteins that were ≥2.0-fold enriched and assigned as synaptic proteins in SynGO (*Figure 2B*, *Figure 3—figure supplement 1B*, *Figure 3—figure supplement 2B*, 'synaptic'), and analyzed each dataset (*Figure 5A–C*). In the control dataset, the proteins formed an integrated network with functional nodes classified as active zone proteins, synaptic vesicle proteins, Ca[2+] regulatory proteins, and synaptic ribosomes (*Figure 5A*). The same nodes were detected in the Syt-1 cKO[DA] mice (*Figure 5C*), substantiating the overall similarities of these datasets. In contrast, these networks were disrupted in RIM cKO[DA] mice (*Figure 5B*), even when hits from the RIM[PPCP]-BirA bait (lighter colors in *Figure 5B*) were included. Only the synaptic ribosome node remained, suggesting that while release site disruption was strong in RIM cKO[DA] mice, other protein complexes near the bait proteins remained intact.

## Disrupted recruitment of α-synuclein to release sites in dopamine axons after impairing dopamine release

Loss of dopamine neurons underlies the motor symptoms characteristic of Parkinson's disease (*Poewe et al., 2017*), and recent human genetic studies establish that mutations in *RIMS* genes increase the risk of Parkinson's disease and are key drivers of disease progression (*Liu et al., 2021b*; *Nalls et al., 2019*). Monogenic forms of Parkinson's disease exist and genetic associations with variable penetrance have been described (*Blauwendraat et al., 2020*; *Day and Mullin, 2021*; *Marras et al., 2016*). Our control dataset contained five of these genes: *Snca, Park7, Dnajc6, Synj-1*, and *Vps35* (*Figure 6A*). Three were enriched in the release site proteome with at least one of the BirA baits, and *Snca*, which encodes the synaptic vesicle associated protein α-synuclein, was ≥2.0-fold enriched in each condition (*Figures 2A and 6A*). In both RIM cKO[DA] and Syt-1 cKO[DA] mice, α-synuclein enrichment was reduced and fell below the 2.0-fold threshold (*Figure 6B*, *Figure 3—figure supplements 1A and 2A*). α-Synuclein is associated with synaptic vesicles and regulates synaptic vesicle clustering, SNARE complex formation, and endocytosis (*Burré et al., 2010*; *Diao et al., 2013*; *Murphy et al., 2000*). RIM or Syt-1 knockout causes a loss of docked vesicles at synapses (*Chang et al., 2018*; *Kaeser et al., 2011*). If these proteins have similar roles in dopamine axons, the loss of α-synuclein enrichment might be explained by reduced vesicle docking.

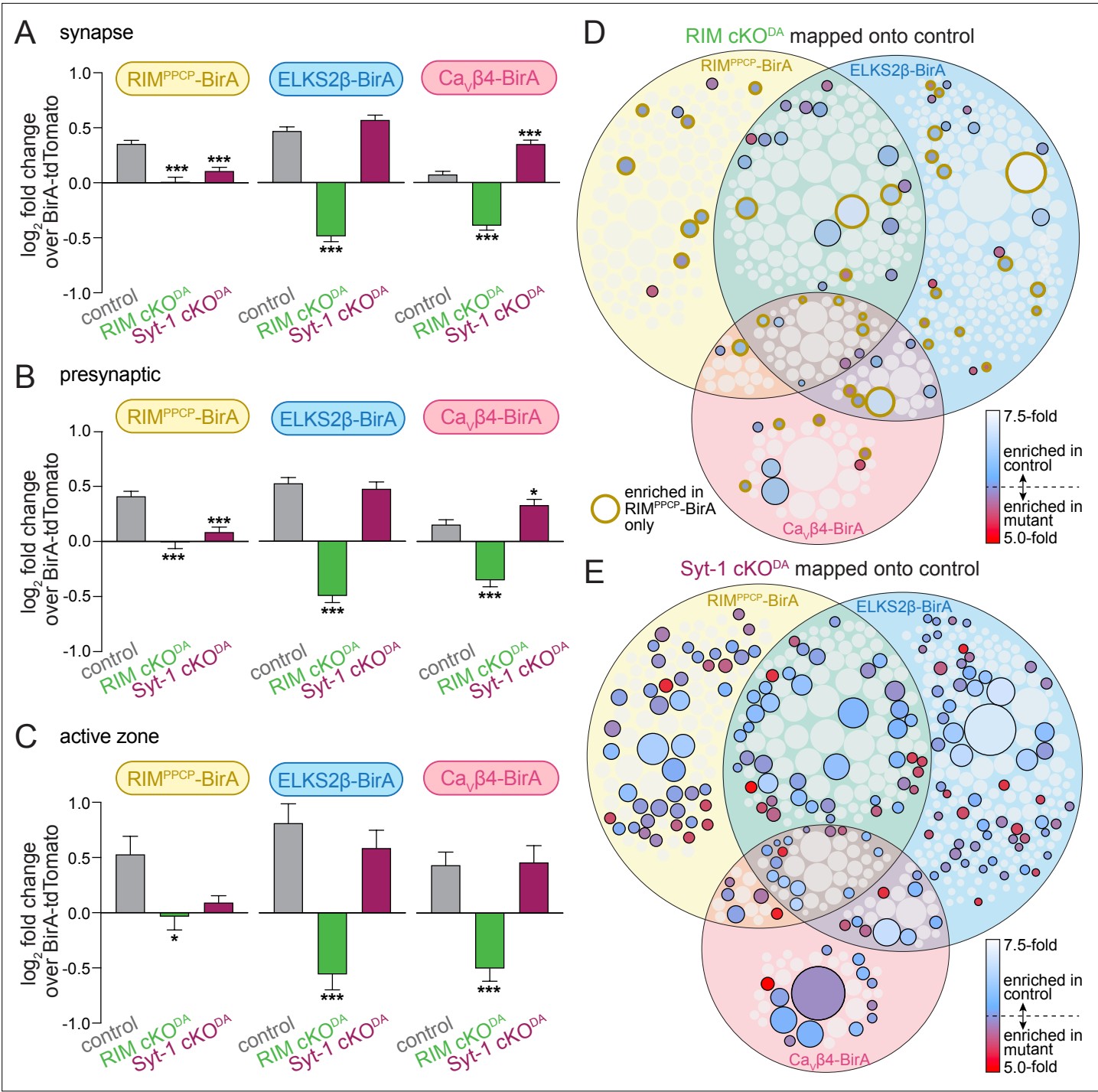

**Figure 4.** RIM cKO[DA] disrupts the protein composition of release sites in dopamine axons. (**A**) The average log$_2$ fold change of identified proteins over the same proteins in the BirA-tdTomato conditions, all proteins with a synaptic localization annotation in SynGO for each bait and genotype are included. Positive values represent release site enrichment and negative values represent depletion relative to axonal protein content assessed with BirA-tdTomato. Total number of proteins detected with a SynGO annotation: control, 540; RIM cKO[DA], 412; Syt-1 cKO[DA], 438. (**B**) Same as (**A**), but for proteins with a presynaptic SynGO annotation. Total number of proteins: control, 257; RIM cKO[DA], 208; Syt-1 cKO[DA], 213. (**C**) Same as (**A**), but for proteins with an active zone SynGO annotation. Total number of proteins: control, 42; RIM cKO[DA], 37; Syt-1 cKO[DA], 36. (**D**) Proteins enriched in the RIM cKO[DA] dataset mapped onto the control Venn diagram shown in *Figure 2A*. Proteins present in the control proteome but not in the RIM cKO[DA] dataset are shown in light gray. Enriched proteins are colored based on relative enrichment (key on the right). Proteins in the RIM cKO[DA] dataset that are only enriched with RIM[PPCP]-BirA are outlined in dark yellow independent of which bait enriched them in the control dataset. (**E**) Same as (**D**), but for the Syt-1 cKO[DA] dataset and without outlining a specific condition. Data in (**A–C**) are shown as mean ± SEM, and significance is presented as *p<0.05, **p<0.01,

*Figure 4 continued on next page*

*Figure 4 continued*

and \*\*\*p<0.001. Two-way ANOVA was used in (**A–C**) (**A**: genotype \*\*\*, bait \*\*\*, interaction \*\*\*; **B**: genotype \*\*\*, bait \*\*, interaction \*\*\*; **C**: genotype \*\*\*, bait \*, interaction not significant), and Bonferroni post-hoc tests (p-values indicated in figure) were used to compare each genotype to control for each BirA bait. For analyses as in (**A–C**) but for proteins with a synaptic vesicle or postsynaptic SynGO annotation, see *Figure 4—figure supplement 1*.

The online version of this article includes the following figure supplement(s) for figure 4:

**Figure supplement 1.** Analyses of enrichment of synaptic vesicle and postsynaptic proteins across genotypes.

## Discussion

We used iBioID to assess release site composition in striatal dopamine axons using three bait proteins. Applying a ≥2.0-fold enrichment threshold, we identified 527 proteins, of which 190 were enriched with multiple baits. This proteome contains known secretory machinery, including active zone, $Ca^{2+}$ regulatory, and synaptic vesicle proteins, and additional proteins not previously associated with vesicular exocytosis. Dopamine neuron-specific knockout of RIM, but not of Syt-1, strongly decreased the number of enriched proteins. We conclude that dopamine axons contain active zone-like sites with secretory machinery for rapid vesicular exocytosis and present an assessment of the protein composition of these sites. Each newly detected protein will require validation for a definitive assignment to dopamine release sites. Our findings establish that the integrity of these active zone-like sites strongly depends on the scaffolding protein RIM.

### Proximity proteomics establish a scaffolding role for RIM in dopamine axons

In previous studies, we found that several active zone proteins are present in release site-like structures of striatal dopamine axons, including Bassoon, RIM, Munc13, and ELKS. Using dopamine-neuron-specific knockout, we established that RIM and Munc13 are essential for evoked axonal dopamine release, while ELKS and RIM-BP are dispensable. RIM removal also induced deficits in Bassoon clustering in striatal dopamine axons. We proposed the model of active zone-like release sites for action potential-evoked dopamine release (*Banerjee et al., 2022*; *Liu et al., 2021a*; *Liu et al., 2018*; *Liu and Kaeser, 2019*). We here used iBioID to generate a list of putative components of these sites. We found that active zone proteins, synaptic vesicle proteins, $Ca^{2+}$ regulatory proteins, and many additional proteins are present. RIM was enriched across datasets, perhaps indicating a role for RIM in dopamine active zone assembly. RIM ablation from dopamine axons indeed resulted in a strong reduction in the number of proteins identified as release site components with iBioID. This loss of material was observed more strongly with ELKS2β-BirA and $Ca_V β4$-BirA than with RIM$^{PPCP}$-BirA, suggesting that RIM$^{PPCP}$-BirA partially restored release site composition. Together with previous data, these findings establish a key scaffolding role for RIM at dopamine release sites. The overall reduced number of hits in RIM cKO$^{DA}$ mice also suggests that many hits in the control proteome are specific.

Reductions in hit numbers in RIM cKO$^{DA}$ mice might at least in part arise from mislocalized BirA baits in their dopamine axons. However, aberrant bait localization likely does not explain all effects. Twenty-two proteins were enriched with all three baits in RIM cKO$^{DA}$ mice, indicating overlapping labeling radii with a resulting proteome relying on bait colocalization (*Figure 3—figure supplement 1A*, *Figure 5*). In a previous study, overall dopamine axon morphology was not strongly disrupted in RIM cKO$^{DA}$ mice apart from changes in Bassoon clustering (*Liu et al., 2018*). Here, the general axonal proteome identified by BirA-tdTomato was similar between RIM cKO$^{DA}$ and control mice (*Figure 3B*), arguing for a generally intact axonal arbor in these mice. Together, these points make it unlikely that bait mislocalization fully accounts for the disruption observed in RIM cKO$^{DA}$ mice. Even if some of the bait protein is mislocalized, the main conclusion that RIM ablation disrupts release site structure in dopamine axons is supported through this mislocalization.

### An active zone-like site for dopamine release

Two recent studies have assessed the composition of dopamine axons or of dopamine varicosities using different methodologies (*Hobson et al., 2022*; *Paget-Blanc et al., 2022*). Neither of the studies was designed to enrich for release site proteins over other proteins in dopamine axons. One used mass spectrometry to analyze dopaminergic synaptosomes obtained through fluorescent sorting (*Paget-Blanc et al., 2022*) and identified 57 proteins associated with these synaptosomes, which contain release

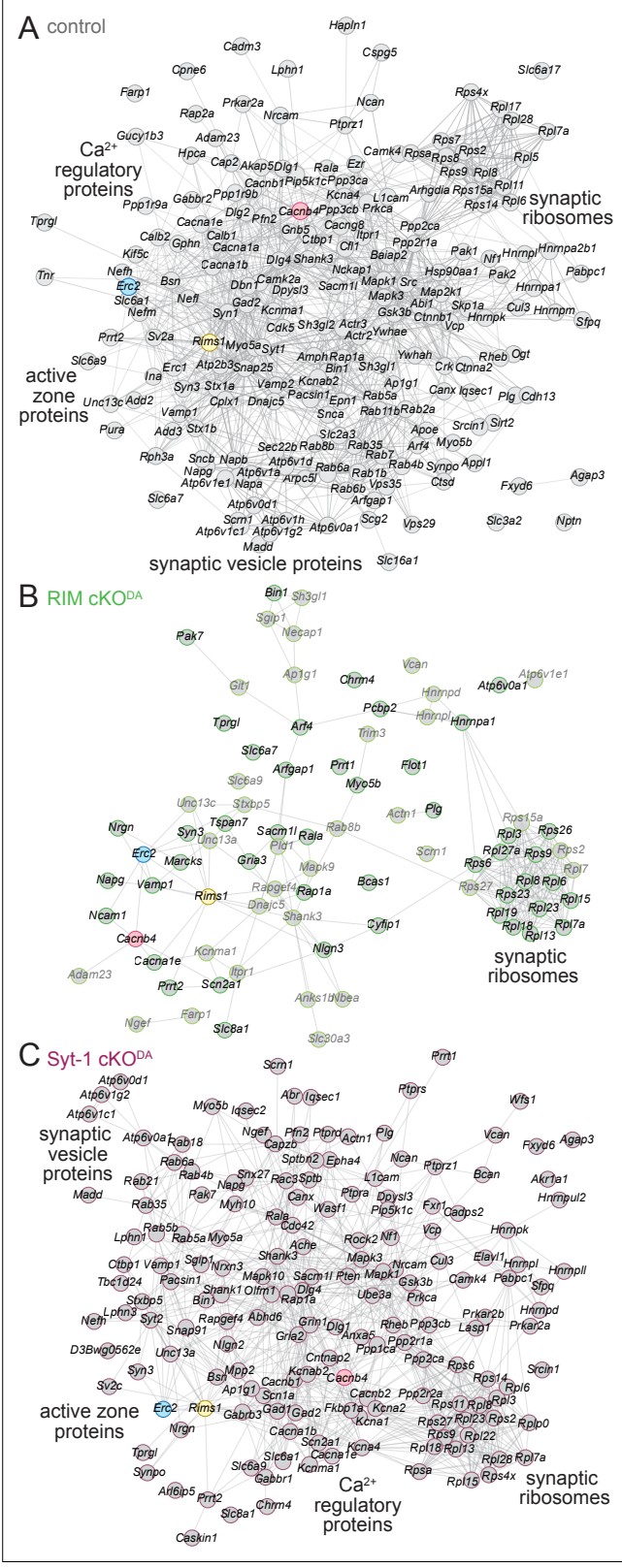

**Figure 5.** STRING diagrams illustrate functional categories of release site proteins and their disruption in RIM cKO[DA] mice. (**A**) STRING diagram of the enriched proteins that have a synaptic SynGO annotation in the control dataset. Physical or functional interactions determined by either empirical data or predictive modeling are illustrated as lines between proteins (*Snel et al., 2000*; *Szklarczyk et al., 2019*). The BirA bait proteins are color-

*Figure 5 continued on next page*

*Figure 5 continued*

coded. Functional categories identified by STRING analyses are labeled close to the corresponding clusters. (**B**) Same as (**A**), but for the RIM cKO[DA] dataset. The proteins that are enriched only in the RIM[PPCP]-BirA bait are shown in lighter gray. (**C**) Same as (**A**), but for the Syt-1 cKO[DA] dataset. Number of proteins: (**A**) 194; (**B**) 79; (**C**) 163.

sites, vesicles, mitochondria, cytoskeletal elements, and many other components. A second study used proximity proteomics with APEX2 to identify axonal proteins (*Hobson et al., 2022*), similar to the BirA-tdTomato condition used for normalization in our experiments. Indeed, there is good overlap between the axonal proteins identified with BirA-tdTomato in our experiments with those in *Hobson et al., 2022*, with 65% of the proteins identified here also present with the APEX2-based method.

We here used an approach to determine enrichment at release sites over these axonal proteomes. RIM1 (*Rims1*), Bassoon (*Bsn*), and ELKS1 (*Erc1*), ELKS2 (*Erc2*), and P/Q-type (Ca$_V$2.1, *Cacna1a*) and N-type Ca$^{2+}$ channels (Ca$_V$2.2, *Cacna1b*) were all detected, consistent with previous studies that assessed roles and/or localization of these proteins in dopamine neurons (*Banerjee et al., 2022*; *Brimblecombe et al., 2015*; *Daniel et al., 2009*; *Ducrot et al., 2021*; *Liu et al., 2022*; *Liu et al., 2018*; *Uchigashima et al., 2016*), and overlapping with a proteomic study that determined release site composition at synapses via purifying proteins associated with docked synaptic vesicles (*Boyken et al., 2013*). The active zone proteins Liprin-α and RIM-BP were not present in the 2.0-fold enriched dataset. We found that Liprin-α2 and -α3 knockout has less severe effects on dopamine release compared to RIM knockout, while removal of RIM-BP leaves dopamine release unimpaired (*Banerjee et al., 2022*; *Liu et al., 2018*). Thus, the presence of active zone proteins in iBioID generally correlates with their known functional roles in dopamine release. One exception to this correlation is Munc13. Striatal dopamine release is strongly impaired after Munc13 ablation (*Banerjee et al., 2022*), but only enriched in some cases with iBioID (*Figures 2 and 5*, *Figure 3—figure supplements 1 and 2*). At synapses, some Munc13 may be more broadly distributed than just at active zones (*Tan et al., 2022a*; *Tan et al., 2022b*), but this does not explain its nonenrichment here because in the BirA-tdTomato dataset, Munc13 was not strongly detected either. Hence, Munc13 may either be overall sparse despite its requirement for evoked dopamine release or may be difficult to detect, for example, because it is poorly biotinylated or not easily affinity-purified. Notably, Munc13 was also absent in the release site proteome assessed via docked vesicles (*Boyken et al., 2013*). Overall, while a saturated proteome is challenging to obtain, many of the proteins previously shown to mediate dopamine release were detected with iBioID.

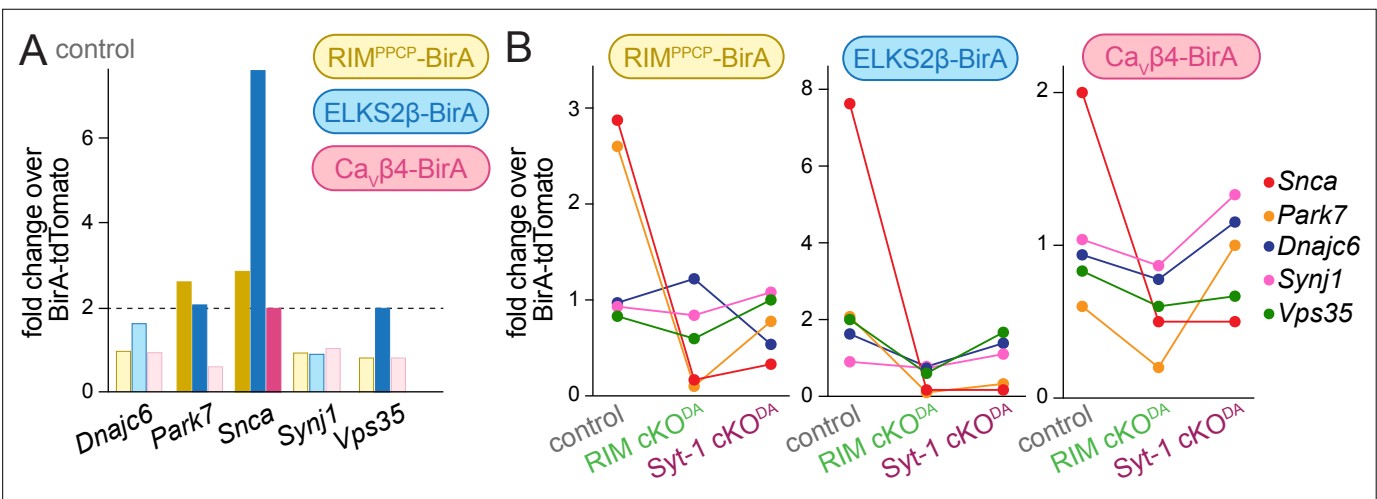

**Figure 6.** Genes associated with Parkinson's disease in the dopamine release site proteomes. (**A**) Enrichment of proteins associated with monogenic forms of Parkinson's disease (*Blauwendraat et al., 2020*; *Day and Mullin, 2021*; *Marras et al., 2016*) in the control dataset. Hits above the 2.0-fold enrichment threshold are shown in saturated colors, and those below the threshold in light colors; control, 4 independent repeats per bait condition. (**B**) Enrichment of the five proteins shown for each bait and in control, RIM cKO[DA] and Syt-1 cKO[DA]; control, 4 independent repeats (as described in *Figure 1*), RIM cKO[DA], 2 (as described in *Figure 3*); Syt-1 cKO[DA], 2 (as described in *Figure 3*).

Dopamine neurons corelease glutamate and GABA (*Hnasko et al., 2010*; *Stuber et al., 2010*; *Tritsch et al., 2012*). While GABA is loaded into vesicles via VMAT2 (*Melani and Tritsch, 2022*; *Tritsch et al., 2012*), glutamate corelease may or may not originate from the same vesicular compartment (*Hnasko et al., 2010*; *Silm et al., 2019*; *Zhang et al., 2015*) and it is currently not possible to determine whether some iBioID hits might be from dedicated release sites for glutamate. Overall, however, our results fit with candidate approaches on dopamine release (*Banerjee et al., 2022*; *Liu et al., 2018*), suggesting that the proteome we describe might in general represent dopamine release sites.

## Postsynaptic, unknown, and unexpected proteins in the dopamine release site proteome

There were 87 proteins with postsynaptic SynGO annotations in the release site proteome. Apart from experimental noise, multiple possibilities may account for this. First, SynGO may not have enough empirical evidence to designate any given protein as presynaptic. Some proteins with only postsynaptic annotations may be genuinely present in the dopamine release machinery. Second, dopamine axons receive input from cholinergic interneurons (*Cachope et al., 2012*; *Threlfell et al., 2012*; *Zhou et al., 2001*), and activation of nicotinic receptors triggers dopamine axon action potential firing (*Liu et al., 2022*). The structure and organization of this cholinergic input are commonly considered nonsynaptic (*Chang, 1988*; *Jones et al., 2001*), but synaptic-like transmission exists (*Kramer et al., 2022*). Hence, dopamine varicosities may in principle contain postsynaptic scaffolds, although nicotinic acetylcholine receptors were not detected with iBioID. Third, the estimated labeling radius of iBioID is up to 50 nm (*Kim et al., 2014*; *Roux et al., 2012*). Striatal dopamine axons make sparse synaptic-like connections with medium spiny neurons that contain the GABAergic postsynaptic scaffold Gephyrin (*Uchigashima et al., 2016*; *Wildenberg et al., 2021*). Gephyrin (*Gphn*) was indeed enriched with ELKS2β-BirA and RIM$^{PPCP}$-BirA (*Figure 2*). Biotinylation across membranes may be common in iBioID, and detection of intramitochondrial proteins and of synaptic vesicle proteins with postsynaptic baits has been observed (*Uezu et al., 2016*). It is noteworthy that in our experiments dopamine receptors were not enriched, consistent with their localization outside of the postsynaptic scaffolds in target neurons (*Uchigashima et al., 2016*).

The proteome presented in *Figure 2* contains 63% proteins that are not annotated in SynGO. Some of these proteins may be real hits that are not included in SynGO, while others may be experimental noise. Cend1, for example, a protein associated with cell cycle exit, appears both in our data and a previous vesicle docking proteome, and electron microscopic studies suggested that it might be present at the presynaptic plasma membrane (*Boyken et al., 2013*; *Patsavoudi et al., 1995*; *Politis et al., 2007*). It is surprising that *Hist1h1d* and *Hist1h4a*, genes encoding for the histone proteins H1.3 and H4, were robustly enriched (*Figure 2A*). These hits might be entirely unspecific or their co-purification could be due to biotinylation of H1 and H4 proteins (*Stanley et al., 2001*). It is also possible that there are unidentified synaptic functions of some of the unexpected proteins. An example for recently identified synaptic functions of nuclear proteins are those of the kinetochore complex (*Zhao et al., 2019*).

Some hits might arise because the BirA baits are not exclusively at release sites. Localization away from release sites is in part expected because a fraction of any release site protein will be away from its endogenous target localization. It might also be caused or amplified by the approach, for example, through viral bait expression and/or modification of protein sequences through bait design. The baits are made in the soma and transported along the axon. Some hits could reflect activity during protein trafficking that occurs within the 7-day time window for biotinylation. Transport complexes may be distinct between the baits and BirA-tdTomato and thus show as enriched. Indeed, several kinesins (*Kif21a*, *Kif21b*, *Kif5a*, *Kif5b*) were enriched, possibly reflecting transport packets. The synaptic ribosomes (*Figure 5*) that reached the 2.0-fold threshold across conditions may also reflect hits by non-active zone-localized bait proteins.

## Summary of conclusions and limitations

Altogether, we present an assessment of dopamine release site content. Each protein will require validation through morphological and functional characterization before an unequivocal assignment to dopamine release sites is possible. As discussed, some hits may be noise, viral expression of tagged and modified bait proteins may influence the results, and thresholding strongly affects the overall number

of proteins included in the release site proteome. We note that iBioID does not identify direct interaction partners. Instead, the defining property is proximity to the bait. The estimated ~50 nm labeling radius goes beyond the radius of protein interactions for most proteins. Despite these limitations, we have made progress. The presented data support a model of active zone-like dopamine release sites for rapid and efficient dopamine secretion. The disruption of the iBioID-generated proteome when RIM is removed establishes a scaffolding role for RIM. This role appears more pronounced in dopamine axons compared to classical synapses. Although similar proteomic approaches were not performed, RIM ablation does not severely disrupt release site structure at classical synapses when assessed with microscopy, but redundant scaffolds might maintain the sites instead (*Acuna et al., 2016*; *de Jong et al., 2018*; *Emperador-Melero and Kaeser, 2020*; *Kaeser et al., 2011*; *Kushibiki et al., 2019*; *Oh et al., 2021*; *Tan et al., 2022a*; *Tan et al., 2022b*; *Wang et al., 2016*). Overall, our work supports the model that dopamine release site scaffolding may have less redundancy and rely heavily on RIM (*Banerjee et al., 2022*; *Liu et al., 2018*).

# Materials and methods

**Key resources table**

| Reagent type (species) or resource | Designation | Source or reference | Identifiers | Additional information |
|---|---|---|---|---|
| Genetic reagent (*Mus musculus*) | B6.SJL-Slc6a3$^{tm1.1(Cre)Bkmm}$/J (DAT$^{IRES-Cre}$) | *Bäckman et al., 2006* | RRID:IMSR_JAX:006660 | |
| Genetic reagent (*M. musculus*) | Rims1$^{tm3Sud}$/J (RIM1αβ$^{fl/fl}$) | *Kaeser et al., 2008* | RRID:IMSR_JAX:015832 | |
| Genetic reagent (*M. musculus*) | Rims2$^{tm1.1Sud}$/J (RIM2αβγ$^{fl/fl}$) | *Kaeser et al., 2011* | RRID:IMSR_JAX:015833 | |
| Genetic reagent (*M. musculus*) | C57BL/6Ntac-Syt1$^{tm1a(EUCOMM)Wtsi}$/WtsiCnrm (Syt-1$^{fl/fl}$) | *Zhou et al., 2015* | RRID:IMSR_EM:06829 | The identifier refers to the line before flp recombination |
| Cell line (*Homo sapiens*) | HEK293T cells | ATCC | Cat#: CRL-3216; RRID:CVCL_0063 | |
| Recombinant DNA reagent | pAAV2/5-syn-DIO-RIM-PRM-PDZ-PxxP-C2A-BirA | This study | LK18005; lab plasmid code (LPC) p864 | This reagent was used to generate AAV viruses and can be obtained from the corresponding author |
| Recombinant DNA reagent | pAAV2/5-syn-DIO- ELKS2β-BirA | This study | LK17008; LPC p857 | This reagent was used to generate AAV viruses and can be obtained from the corresponding author |
| Recombinant DNA reagent | AAV2/5-syn-DIO-Ca$_V$β4-BirA | This study | LK19004; LPC p868 | This reagent was used to generate AAV viruses and can be obtained from the corresponding author |
| Recombinant DNA reagent | AAV2/5-syn-DIO-BirA-tdTomato | This study | LK17011; LPC p860 | This reagent was used to generate AAV viruses and can be obtained from the corresponding author |
| Antibody | Anti-HA (rabbit polyclonal) | Cell Signaling Technology | CAT# 5017; RRID:AB_10693385, lab antibody code (LAC) A40 | Immunofluorescence (IF) (1:500) |
| Antibody | Anti-HA (mouse monoclonal) | BioLegend | CAT# 901501; RRID:AB_2565006, LAC A12 | IF (1:500) Western blot (WB) 1:500 |
| Antibody | Anti-tyrosine hydroxylase (guinea pig polyclonal) | SySy | CAT# 213 104; RRID:AB_2619897, LAC A111 | IF (1:1000) |
| Antibody | Anti-Bassoon (mouse monoclonal) | Millipore | CAT# ADI-VAM-PS003-F; RRID:AB_11181058, LAC A85 | IF (1:500) |
| Antibody | Anti-Neuroplastin (goat polyclonal) | R&D Systems | CAT# AF5360; RRID:AB_2155920, LAC A253 | IF (1:500) |

*Continued on next page*

*Continued*

| Reagent type (species) or resource | Designation | Source or reference | Identifiers | Additional information |
|---|---|---|---|---|
| Antibody | Anti-red fluorescent protein (rabbit polyclonal) | Rockland | CAT#600-401-379; RRID:AB_2209751, LAC A81 | WB (1:1000) |
| Antibody | Anti-β-actin (mouse monoclonal) | Sigma-Aldrich | CAT# A1978; RRID:AB_476692, LAC A127 | WB (1:10,000) |
| Antibody | Anti-pyruvate carboxylase (rabbit polyclonal) | Novus | CAT# NBP1-49536G; RRID:AB_11016707, LAC A252 | This antibody was used for depletion as described in the 'Materials and methods' |
| Software, algorithm | Prism9 | GraphPad | RRID:SCR002798; https://www.graphpad.com/scientific-software/prism | |
| Software, algorithm | Cytoscape v 3.8.2 | Cytoscape | RRID:SCR003032; http://cytoscape.org/ | |
| Software, algorithm | SEQUEST PRO | Thermo Fisher | https://scicrunch.org/resolver/SCR_014594 | |
| Software, algorithm | MATLAB code for object analysis | *Liu, 2021* | https://github.com/kaeserlab/3DSIM_Analysis_CL | |

## Mouse lines

Expression via AAVs and ablation of active zone proteins in dopamine neurons was performed in mice with Cre recombinase specifically expressed in these neurons (*Bäckman et al., 2006*) (Jackson Laboratories; RRID:IMSR_JAX:006660, *B6.SJL-Slc6a3$^{tm1.1(Cre)Bkmm}$*/J, also called DAT$^{IRES-Cre}$ mice). iBioID to generate the control proteome was performed in mice heterozygote for this allele. Conditional ablation of RIM in dopamine neurons (RIM cKO$^{DA}$) was performed as described (*Liu et al., 2018*). Mice with floxed alleles for *Rims1* (to remove RIM1α and RIM1β, RRID:IMSR_JAX:015832, *Rims1$^{tm-3Sud}$*/J) (*Kaeser et al., 2008*) and *Rims2* (to remove RIM2α, RIM2β and RIM2γ, RRID:IMSR_JAX:015833, *Rims2$^{tm1.1Su}$d*/J) (*Kaeser et al., 2011*) were crossed to DAT$^{IRES-Cre}$ mice. Conditional ablation of Syt-1 in dopamine neurons (Syt-1 cKO$^{DA}$) was performed as described (*Banerjee et al., 2020*). Mice with floxed alleles for *Syt-1* (RRID:IMSR_EM:06829, *C57BL/6Ntac-Syt1$^{tm1a(EUCOMM)Wtsi}$*/WtsiCnrm) (*Skarnes et al., 2011*) were flp-recombined and analyzed before (*Kochubey et al., 2016*; *Zhou et al., 2015*). They were crossed to DAT$^{IRES-Cre}$ mice as described in *Banerjee et al., 2020*. For generating cohorts of mice for iBioID, the floxed alleles were homozygote in both parents and one parent contained a heterozygote DAT$^{IRES-Cre}$ allele. Male and female mice were used in all experiments irrespective of sex. All animal experiments were approved by the Harvard University Animal Care and Use Committee (protocol number IS00000049).

## Plasmids, cell lines, and AAVs

AAVs were generated to express BirA fusion proteins in dopamine neurons. The following cDNAs were produced by standard techniques (for sequences, see source data tables): LK18005-AAV2/5-syn-DIO-RIM$^{PPCP}$-BirA (also called LK18005-AAV2/5-syn-DIO-RIM-PRM-PDZ-PxxP-C2A-BirA, p864), LK17008-AAV2/5-syn-DIO-ELKS2β-BirA (p857), LK19004-AAV2/5-syn-DIO-Ca$_V$β4-BirA (p868), and LK17011-AAV2/5-syn-DIO-BirA-tdTomato (p860). A cDNA encoding a promiscuous version of BirA (also called BirA*) was provided by S. Soderling and A. Uezu (*Roux et al., 2012*; *Uezu et al., 2016*). p864 to express RIM$^{PPCP}$-BirA was generated based on a RIM full-length cDNA (*de Jong et al., 2018*; *Kaeser et al., 2011*; *Tan et al., 2022b*), and it contained endogenous proline-rich motifs (PRMs, also called PxxP motifs), the PDZ and C2A domains, and corresponding endogenous linkers, with BirA flanked by HA-tags added at the C-terminus. The RIM$^{PPCP}$-BirA protein had the following sequence (numbered in subscript according to amino acid sequences of XM_039084276.1 for RIM and of UVH36278.1 for BirA): M-$_{RIM1,524}$RPSP...GSIEQ$_{1120}$-AAAYPYDVPDYA-$_{BirA,154}$DNTV...SAEK$_{472}$-AYPYDVPDYA. p857 to express ELKS2β-BirA was produced based on previously published N-terminally tagged ELKS2β (*Kaeser et al., 2009*) with removal of most of the N-terminal tag and insertion of BirA flanked by

HA sequences between the coiled-coil regions $CC_C$ and $CC_D$. The ELKS2β-BirA protein had the following sequence (numbered in subscript according to amino acid sequences of NM_170787.3 for ELKS2 and of UVH36278.1 for BirA): MGAALNRSVQTCFCSRMPCEQQICSH-$_{ELKS2,367}$ELHRR…NIED$_{657}$-AAAYPYDVPDYA-$_{BirA,154}$DNTV….SAEK$_{472}$-AYPYDVPDYAISRWRA-$_{ELKS2,658}$DSRMN….GIWA$_{921}$. p868 to express $Ca_V$β4-BirA was produced from a previously used plasmid that was generated from a cDNA obtained from Annette Dolphin through Addgene (Addgene #107426; https://www.addgene.org/107426/; RRID:Addgene_107426; *Brodbeck et al., 2002*; *Tan et al., 2022b*) with addition of a linker sequence and an HA-flanked BirA protein at the C-terminus. The $Ca_V$β4-BirA protein had the following sequence (numbered in subscript according to amino acid sequences of NM_001399143.1 for $Ca_V$β4 and of UVH36278.1 for BirA): $_{CaVβ4,1}$MSSS...RHRL$_{519}$-VYNPAHNIEDAAAYPYDVPDYA-$_{BirA,154}$DNTV...SAEK$_{472}$-AYPYDVPDYA. p860 to express BirA-tdTomato was generated by fusing the BirA protein with tdTomato connected with a linker sequence that included an HA-tag. The BirA-tdTomato protein had the following sequence (numbered in subscript according to amino acid sequences of UVH36278.1 for BirA and of AJP62580.1 for tdTomato): M-$_{BirA,154}$DNTV...SAEK$_{472}$-AYPYDVPDYAGAPAS-$_{tdTomtato,2}$VSKG...ELYK$_{476}$. AAVs were made in HEK293T cells (purchased mycoplasma free from ATCC, CRL-3216, RRID:CVCL_0063, immortalized human cell line of female origin) using $Ca^{2+}$ phosphate transfection and were of the serotype 2/5. Three days after transfection, HEK293T cells were collected and stored in freezing buffer (150 mM NaCl, 20 mM Tris-Cl, 2 mM $MgCl_2$, pH 8.0) at –20°C until the virus was purified. For purification of AAVs, cells were lysed by three freeze-thaw cycles with dry ice/ethanol and a 37°C incubator. After 1 hr benzonase nuclease treatment at 37°C, cells were loaded onto an iodixanol gradient (5 ml each, 15%, 25%, 40%, 60%) and ultra-centrifuged at 208,000 × $g$ for 4 hr. Viral particles were then purified from the 40% layer of the gradient. Quantitative reverse transcriptase PCR was used to determine viral titers, and viruses were used at concentrations ranging from $4.0 \times 10^{11}$ to $9.6 \times 10^{12}$ viral genome copies/ml.

## Stereotaxic surgery and biotin injections

Mice (at postnatal days 30–55) were anesthetized in a 5% isoflurane induction chamber and then mounted on a stereotaxic frame; anesthesia was maintained with 1.5–2% isoflurane for the length of the surgery with a nose cone. The scalp was cut open and a hole was drilled in the skull and 1 µl of AAV viral solution was injected in the substantia nigra pars compacta (right or bilaterally depending on the experiment, 0.6 mm anterior, ±1.3 mm lateral of Lambda and 4.2 mm below the surface of the brain) using a microinjector pump (PHD ULTRA, Harvard Apparatus) at 100 nl/min. Mice were treated with post-surgical analgesia and were allowed to recover for at least 28 days prior to biotin injections (for iBioID) or transcardial perfusion (for morphological experiments). Biotin injections were started 4–6 weeks after stereotaxic AAV injection. Mice were subcutaneously injected for seven consecutive days with 500 µl of 5 mM biotin dissolved in phosphate-buffered saline (PBS).

## Immunostaining and confocal imaging of brain sections

At least 4 weeks after stereotaxic injection of BirA viruses, mice (58–100 days old) were deeply anesthetized with isoflurane. Transcardial perfusion was performed with 30–50 ml ice-cold PBS followed by 50 ml of 4% paraformaldehyde in PBS (4% PFA) at 4°C. Brains were then dissected out and incubated in 4% PFA for overnight at 4°C. Fixed brains were sliced on a vibratome (Leica, VT1000s) at 100 µm thickness. Coronal sections containing the midbrain and striatum were collected in ice-cold PBS. Sections were blocked in PBS containing 0.25% Triton X-100 and 10% goat serum (PBST) for 1 hr at room temperature. Slices were incubated overnight in primary antibody in PBST at 4°C. The following primary antibodies were used: rabbit polyclonal anti-HA (1:500, A40, RRID:AB_10693385) or mouse monoclonal anti-HA (1:500, A12, RRID:AB_2565006), and guinea pig polyclonal anti-TH (1:1000, A111, RRID:AB_2619897). Slices were washed three times in PBST followed by 2 hr incubation with secondary antibody in PBST for 2 hr at room temperature in the dark. The following secondary antibodies were used: goat anti-rabbit IgG Alexa 488 (1:500, S5, RRID:AB_2576217), goat anti-mouse IgG1 Alexa 488 (1:500, S7, RRID:AB_2535764), and goat anti-guinea pig IgG Alexa 633 (1:500, S34, RRID:AB_2535757). Slices were washed again three times with PBST before being mounted on glass slides with Fluoromount-G (Southern Biotech 0100-01). Stained slices were then imaged on an Olympus FV1000 confocal microscope with a 60× objective. Images were pseudo-colored in ImageJ for display. All image acquisition was done in comparison to an uninfected control

imaged at the same time to assess background fluorescence. Representative images were brightness and contrast adjusted to facilitate inspection, and these adjustments were made identically for images within the same experiment. All images in *Figure 1C* except for those of mice expressing ELKS2β-BirA were taken at the same time. Images of mice expressing ELKS2β-BirA were taken in a separate session and compared to their own uninfected control to confirm signal specificity (*Figure 1—figure supplement 1*). Three to five images per virus condition were taken during each imaging session, and the experiment was repeated in at least three mice per condition.

## Biotin affinity purification

Biotin affinity purifications were adapted from previously established methods (*Uezu et al., 2016*), 2 (for pilot experiments, up to 4) to 12 mice (for mass spectrometry analyses) were used per experiment and condition. The mice were 65–100 days old and previously injected with AAVs for BirA fusion protein expression and subjected to subcutaneous biotin injections. They were deeply anesthetized with isoflurane and decapitated. Brains were collected in ice-cold PBS and striata were dissected out. Except for those used in pilot experiments, striata were then flash frozen and stored at –80°C until further processing. For pilot experiments, 2–4 dissected striata were immediately homogenized. For mass spectrometry, 5–6 dissected striata were homogenized at a time. Homogenization was performed using a glass-Teflon homogenizer in 1 ml of homogenizing buffer (50 mM HEPES, 150 mM NaCl, 1 mM EDTA+ mammalian protease inhibitor cocktail, Sigma Cat# P8340) with 30 slow strokes on ice. An appropriate volume of 5× lysis buffer (1% SDS, 5% Triton X-100, 5% deoxycholate in homogenizing buffer) was added to the homogenized tissue (working concentration 0.2% SDS, 1% Triton X-100, 1% deoxycholate) and incubated while rotating at 4°C for 1 hr. The mass spectrometry samples were split into two batches of 5–6 striata per condition for the initial homogenization, and the batches were combined after lysis into a single tube. Lysed samples were then sonicated twice for 10 s each at 4°C with a Branson Sonifier 450. Sonicated samples were centrifuged at 15,000 × *g* for 15 min at 4°C in a table-top centrifuge. The cleared supernatant was removed and added to open-top polycarbonate tubes (Beckman Cat# 343778), and centrifuged in a table-top ultracentrifuge (Beckman Rotor TLA120.2) for 1 hr at 100,000 × *g*. After ultracentrifugation, SDS from 0.4% and/or 5% SDS stock solutions was added to the cleared supernatant to adjust to an SDS concentration of 1% and a volume of 1.5 ml. The sample was boiled at 95°C for 5 min and allowed to cool to room temperature. Neutravidin agarose beads (Thermo Cat# 29200) were washed three times in binding buffer (1% SDS, 1% Triton X-100, 1% deoxycholate in homogenizing buffer). 20 µl washed neutravidin beads were added to each sample and the samples were incubated for 16 hr at 4°C on a rotator. After spinning at 500 × *g* for 2 min at 4°C, the supernatant was removed and the beads were transferred to Protein Lo-bind tubes (Eppendorf Cat# 022431081). Beads were washed twice with 500 µl 2% SDS in H$_2$O, twice with 500 µl 1% Triton X-100, 1% deoxycholate, 25 mM LiCl in H$_2$O, and twice in 500 µl 1 M NaCl. For each washing step, beads were pelleted by spinning at 500 × *g* for 2 min in a table-top centrifuge at 4°C. Bead pellets were then washed five times in 500 µl 50 mM ammonium bicarbonate in water with spinning at 500 × *g* for 2 min between steps. After the final wash, the bead pellet was stored at –20°C until next steps (either Western blot or PC removal).

## Western blot after biotin pulldown

Neutravidin bead pellets were incubated in 60 µl of 1× SDS-PAGE loading buffer and boiled for 10 min at 95°C. The sample was spun at 13,000 × *g* in a table-top centrifuge for 1 min to pellet the beads. 15 µl of the supernatant were loaded on an SDS-PAGE gel, and 1% of the total input used for the binding reaction (cleared lysate just before addition of the neutravidin beads) was also loaded onto the gel. After gel electrophoresis, proteins were transferred onto a nitrocellulose membrane. Membranes were blocked in 10% milk, 5% goat serum in Tris-buffered saline + 0.1% Tween-20 (TBST) for 1 hr at room temperature and then incubated overnight at 4°C in primary antibody diluted in antibody binding solution (blocking solution diluted 1:1 with TBST). Primary antibodies used: mouse monoclonal anti-HA (1:500, LAC A12, RRID:AB_2565006), rabbit polyclonal anti-red fluorescent protein (RFP) (1: 500, LAC A81, RRID:AB_2209751), rabbit polyclonal anti-pyruvate carboxylase (1:500, LAC A252, RRID:AB_11016707), and mouse monoclonal anti-β-actin (1:10,000, LAC A127, RRID:AB_476692). Membranes were washed three times with TBST and incubated in HRP-conjugated

secondary antibodies for 1 hr at room temperature. Membranes were washed three times in TBST and enhanced chemiluminescence followed by exposure to film was used to visualize protein bands.

## Pyruvate carboxylase depletion

For samples being submitted to mass spectrometry, frozen neutravidin bead pellets were thawed on ice. Proteins were eluted by incubation in 500 µl RapiGest elution buffer (0.1% RapiGest, Waters Cat# 1866001861, in 2 mM biotin, 50 mM ammonium bicarbonate) for 2 hr at 60°C, while shaking. Neutravidin beads were pelleted by spinning at 18,000 × $g$ for 5 min at 4°C and the supernatant was moved to a new tube. To prepare anti-pyruvate carboxylase (PC) antibody-conjugated beads, Protein G Sepharose beads (GE Healthcare Cat#17-0618-01) were washed three times in RapiGest elution buffer and conjugated to anti-PC antibodies (Novus, A252, RRID:AB_11016707) by incubating 3 µl of antibody per 20 µl of beads for 1 hr at 4°C on a rotator. Conjugated beads were spun down at 1000 × $g$ for 2 min, the supernatant was removed, and beads were diluted 1:1 with fresh RapiGest elution buffer to make a 50% slurry. 20 µl sepharose beads (40 µl of a 50% slurry) were added to the 500 µl of eluted protein solution of the biotin affinity purification and incubated for 1 hr at 4°C on a rotator. PC-conjugated beads were pelleted by spinning the sample at 18,000 × $g$ for 5 min. The proteins in the PC-depleted supernatant were then precipitated with trichloroacetic acid (TCA). One volume of 100% TCA was added to four volumes of the PC-depleted supernatant and inverted several times before incubation on ice for 10 min. Tubes were then spun at 20,000 × $g$ for 10 min at 4°C. The supernatant was removed leaving behind a protein pellet. The pellet was air dried and stored at –20°C until analysis by mass spectrometry. In pilot experiments, we assessed the efficiency of PC depletion. Before PC depletion, 28% and 10% of the detected peptides were from PC and the related protein propionyl-CoA carboxylase (PCCA), respectively. After depletion, 3% and 1.7% of the detected peptides were from PC and PCCA, respectively.

## Mass spectrometry

Liquid chromatography tandem mass spectrometry (LC-MS/MS) was performed by the Taplin Mass Spectrometry Facility of the Department of Cell Biology at Harvard Medical School. Either the neutra-vidin bead pellet (some pilot experiments) or the TCA-precipitated protein pellet after PC depletion (all experiments to assess release site composition) were submitted to the facility. The pellet was subjected to 5 ng/µl trypsin digest overnight at 37°C and dried until further analysis. On the day of analysis, the samples were reconstituted in 2.5% acetonitrile, 0.1% formic acid, and loaded via a Famos auto sampler (LC Packings, San Francisco, CA) onto the column after column equilibration. Peptides were eluted with increasing concentrations of 97.5% acetonitrile and 0.1% formic acid. As peptides were eluted, they were subjected to electrospray ionization and added to an LTQ Orbitrap Velos Pro ion-trap mass spectrometer (Thermo Fisher Scientific). Peptides were detected, isolated, and fragmented to produce a tandem mass spectrum of specific fragment ions for each peptide. Peptide sequences (and hence protein identity) were determined by matching protein databases with the acquired fragmentation pattern by the SEQUEST software program (Thermo Fisher Scientific). The data were filtered to establish a false discovery rate between 1% and 2% using a database of mouse protein sequences made up of half real protein sequences and half reversed sequences. Independent of whether the results were run against a mouse or a rat database, the number of peptides identified for bait proteins (which were made from rat cDNA) was similar.

## Analyses of mass spectrometry data

The control dataset consists of four repeats that stem from two separate mass spectrometry sessions (repeats 1 and 2 were run first, repeats 3 and 4 were run ~1 year later). Each mutant dataset (RIM cKO[DA] and Syt-1 cKO[DA]) consists of two repeats that were assessed in the same mass spectrom-etry session together with repeats 3 and 4 of the control dataset. Mitochondrial proteins were removed from the analyses using MitoCarta3.0 (*Pagliarini et al., 2008*; *Rath et al., 2021*), except for *Figures 3B and 6*. PC and PCCA were also removed from the analyses along with neutravidin and IgGs. Peptide counts for each protein in each BirA condition (RIM[PPCP]-BirA, ELKS2β-BirA, Ca$_V$β4-BirA, BirA-tdTomato) were first averaged for the two repeats that were acquired in the same mass spectrometry session. If the peptide count averaged from the two repeats was zero, it was assigned a value of 0.5 such that we could calculate circle size for *Figure 2* and more generally enrichment or

depletion for all analyses. This assignment warranted that for a protein to be included, it had to be detected more than once with a single peptide. The average peptide count for a given protein for each of the three BirA baits was divided by the average peptide count for the same protein in the BirA-tdTomato condition. This is referred to as the fold change (FC) value. For the control dataset, the average fold change values of each mass spectrometry session were then averaged together to generate a final fold change value for every protein shown in *Figure 2* and *Figure 2—figure supplements 1 and 2*; in the mutant datasets, this final averaging was not performed as all data stem from a single mass spectrometry session. Proteins with at least 2.0-fold enrichment over BirA-tdTomato in any release site-BirA condition were considered 'hits' in *Figure 2* and alternate thresholds of 1.5-fold and 2.5-fold enrichment are shown in *Figure 2—figure supplements 1 and 2*. The Venn diagrams were constructed using Cytoscape (v3.8.2) (*Shannon et al., 2003*) to map the circle size to fold change value. Circle size reflects average enrichment across repeats of an individual bait. For proteins enriched in multiple bait conditions, circle size corresponds to the condition with the largest average enrichment value. The $\log_2$ of the fold change was calculated for any protein that had at least a single peptide found for any of the BirA fusion proteins. To sort proteins for synaptic localization, either the enriched proteins (*Figures 2B and 5*, *Figure 2—figure supplement 3*, *Figure 3—figure supplements 1 and 2*) or all proteins (*Figure 4A–C*, *Figure 4—figure supplement 1*) were run through the SynGO database (https://www.syngoportal.org; *Koopmans et al., 2019*) and selected by their cellular component annotation. For *Figure 4A–C* and *Figure 4—figure supplement 1*, the self-biotinylated protein was removed from its own release site-BirA condition (e.g., RIM was removed from analysis in the RIM^PPCP-BirA dataset but not in the Ca_vβ4- or ELKS2β-BirA datasets). For *Figure 5*, the enriched proteins that possessed SynGO annotations were run through the STRING database (v11) (https://string-db.org; *Szklarczyk et al., 2019*) to generate a STRING network diagram of enriched synaptic proteins.

## Synaptosome preparation and staining

Synaptosome experiments were performed according to previously established protocols (*Banerjee et al., 2022*; *Liu et al., 2018*). Mice (30–60 days old) were deeply anesthetized with isoflurane and decapitated. The brain was collected in ice-cold PBS, and the striatum was dissected out. The striatal sections were homogenized in 1 ml of synaptosome homogenizing buffer (320 mM sucrose, 4 mM HEPES, 1× Sigma protease inhibitor cocktail [Cat# P8340], pH 7.4), with 12 slow strokes using a glass-Teflon homogenizer. One ml of homogenizing buffer was then added to the homogenate. The homogenate was spun at 1000 × *g* for 10 min at 4°C, the supernatant was pipetted into a new tube, and spun again at 12,500 × *g* for 15 min at 4°C. The pellet ('P2 fraction') was resuspended in 1 ml homogenizing buffer and homogenized again with six slow strokes. An additional 1 ml of homogenizing buffer was added, and the sample (~1.5 ml) was loaded onto a sucrose gradient made up of 5 ml 1.2 M sucrose on the bottom and 5 ml of 0.8 M sucrose at the top in thin wall ultracentrifugation tubes (Beckman Coulter, Cat# 344059). The loaded gradient was ultracentrifuged at 69,150 × *g* for 1 hr at 4°C (SW 41 Ti Swinging-Bucket Rotor, Beckman Coulter, Cat# 331362) and the synaptosome fraction was harvested from the interface between the two sucrose layers. The synaptosome fraction was diluted 20- to 40-fold in homogenizing buffer and 1 ml was added to a poly-D-lysine-coated coverslip (neuVitro Cat# GG-18-1.5) in a 12-well plate and spun for 15 min at 4000 × *g*. The buffer was removed and the synaptosomes adhering to the coverslips were fixed with 4% PFA in PBS for 10 min. The PFA was removed, and a solution with 3% bovine serum albumin and 0.1% Triton X-100 in PBS (TBP) was used for blocking and permeabilization for 1 hr at room temperature. The following primary antibodies were used (diluted in TBP) overnight at 4°C: goat polyclonal anti-NPTN (1:500, A253, RRID:AB_2155920), mouse monoclonal IgG2a anti-Bassoon (1:1000, A85, RRID:AB_11181058), and guinea pig polyclonal anti-TH (1:1000, A111, RRID:AB_2619897). After primary antibody incubation, the coverslips were washed three times in TBP and then incubated in secondary antibodies for 2 hr at room temperature in the dark. Secondary antibodies used were donkey anti-goat Alexa 488 (1:500, S6, RRID:AB_2534102), donkey anti-mouse Alexa 555 (1:500, S48, RRID:AB_2534013), and donkey anti-guinea pig Alexa 647 (1:500, S59, RRID:AB_2340476). The stained synaptosomes were washed three more times in TBP before mounting on glass slides with Fluoromount-G (Southern Biotech 0100-01).

## Confocal microscopy and image analyses of stained synaptosomes

Coverslips with fixed synaptosomes were imaged with an oil immersion 60× objective and a 1.5× optical zoom on an Olympus FV1000 confocal microscope. Raw confocal images were analyzed in a custom MATLAB program as described before (*Banerjee et al., 2022*; *Liu et al., 2018*) and the code was deposited on GitHub (https://github.com/kaeserlab/3DSIM_Analysis_CL; *Liu, 2021*). 1000–2000 synaptosomes per image were detected using Otsu intensity thresholds and size thresholds (0.2–1 $\mu m^2$ for TH and 0.15–2 $\mu m^2$ for Neuroplastin and Bassoon). These threshold settings were identical for every image analyzed and used to detect Bassoon-positive (Bassoon+) ROIs, TH-positive (TH+) ROIs, and double-positive ROIs (Bassoon+/TH+). TH+ ROIs that had a Bassoon signal less than 1× the average intensity of all pixels in the image were designated as Bassoon-negative (Bassoon-). Neuroplastin antibody signal intensities within Bassoon+ TH+ or Bassoon- TH+ ROIs were quantified and a frequency histogram was plotted. The experimenter was blind to which group a specific particle belonged to during image acquisition and analyses.

## Statistics

Statistics were performed in GraphPad Prism 9. Data are displayed as individual data points, mean ± SEM, and/or frequency histograms. Significance is presented as *$p < 0.05$, **$p < 0.01$, and ***$p < 0.001$. Sample sizes were determined based on previous studies, no statistical methods were used to predetermine sample size, and no outliers were excluded. An unpaired two-tailed Student's *t*-tests was used in *Figure 2—figure supplement 4B*, a Kolmogorov–Smirnov test was used in *Figure 2—figure supplement 4C*, a two-way ANOVA test was used in *Figure 3B*, and two-way ANOVA tests followed by Bonferroni post-hoc tests were used in *Figure 4A–C* and *Figure 4—figure supplement 1*. In all figures, sample sizes and the specific tests used are stated in the figure legends.

## Materials, data and code availability

Plasmids will be shared upon request within the limits of respective material transfer agreements. Mouse mutant alleles are publicly available as outlined in the Key Resources Table. The previously published code used for analyses of synaptosomes has been deposited to GitHub (https://github.com/kaeserlab/3DSIM_Analysis_CL; *Liu, 2021*) and is publicly available as listed in the Key Resources Table. All data generated or analyzed in this study are included in the figures and the source data tables. Source data files are provided for *Figures 1–3*, *Figure 1—figure supplement 2*, *Figure 2—figure supplements 1 and 2*, and *Figure 3—figure supplements 1 and 2*. Requests for materials should be directed to the corresponding author (kaeser@hms.harvard.edu).

## Acknowledgements

We thank J Wang and C Qiao for technical support, current and former members of the Kaeser laboratory and D Brann for insightful discussions and feedback, M Feany and H Nyitrai for comments on the manuscript, R Tomaino and the Taplin Mass Spectrometry Facility of the Department of Cell Biology at Harvard Medical School for help with mass spectrometric analyses, and S Soderling and A Uezu for sharing the BirA plasmid and for advice on iBioID early in the project. LK is currently an employee of Mass General Brigham (Boston, MA, USA). This work was supported in part by the NIH (R01NS103484, R01NS083898 to PSK; F31NS105159 to LK), the Lefler foundation (to PSK), a Dean's Innovation grant (PSK), a Quan fellowship (LK), an Alice and Joseph Brooks fellowship (AB), and Harvard Medical School. We acknowledge the Neurobiology Imaging Facility (supported by a P30 Core Center Grant P30NS072030).

## Additional information

### Funding

| Funder | Grant reference number | Author |
|---|---|---|
| National Institute of Neurological Disorders and Stroke | R01NS103484 | Pascal S Kaeser |
| National Institute of Neurological Disorders and Stroke | R01NS083898 | Pascal S Kaeser |
| National Institute of Neurological Disorders and Stroke | F31NS105159 | Lauren Kershberg |
| Harvard Medical School | Lefler Foundation | Pascal S Kaeser |
| Harvard Medical School | Brooks Fellowship | Aditi Banerjee |
| Harvard Medical School | Quan Fellowship | Lauren Kershberg |
| Harvard Medical School | Dean's Innovation Grant | Pascal S Kaeser |
| Harvard Medical School | | Pascal S Kaeser |

The funders had no role in study design, data collection and interpretation, or the decision to submit the work for publication.

### Author contributions

Lauren Kershberg, Conceptualization, Resources, Formal analysis, Investigation, Methodology, Writing – original draft, Writing – review and editing; Aditi Banerjee, Resources, Formal analysis, Investigation, Methodology, Writing – review and editing; Pascal S Kaeser, Conceptualization, Formal analysis, Supervision, Funding acquisition, Writing – original draft, Writing – review and editing

### Author ORCIDs

Lauren Kershberg http://orcid.org/0000-0002-5676-4160
Aditi Banerjee http://orcid.org/0000-0003-2016-0717
Pascal S Kaeser http://orcid.org/0000-0002-1558-1958

### Ethics

All animal experiments were performed according to institutional guidelines of Harvard University, and were in strict accordance with the recommendations in the Guide for the Care and Use of Laboratory Animals of the National Institutes of Health. All animal experiments were approved by the Harvard University Animal Care and Use Committee (protocol number IS00000049).

### Decision letter and Author response

Decision letter https://doi.org/10.7554/eLife.83018.sa1
Author response https://doi.org/10.7554/eLife.83018.sa2

---

## Additional files

### Supplementary files

• MDAR checklist

• Source data 1. Source data for *Figures 1–3* and figure supplements. (A) Table of proteins detected in mass spectrometry in control mice. An alphabetical list of genes encoding the proteins identified by mass spectrometry and associated peptide counts in the control datasets is provided. (B) DNA sequences. Sequences of the inserts of newly generated plasmids used for producing AAVs are provided. Only the inserts in the multiple cloning site 3′ of the human synapsin promotor of a standard AAV vector are included. (C) Table of proteins for the Venn diagram in *Figure 2*. Genes encoding the bait proteins are shown in the first three rows. The table contains an alphabetical list ordered by enrichment group (enriched in all bait conditions, enriched in two bait conditions, enriched in one bait condition). To be included in the Venn diagram, enriched is defined as an

average ≥2.0-fold number of peptides for a specific protein over the peptide count for the same protein in the BirA-tdTomato condition, calculated across biological repeats and mass spectrometry experiments as described in detail in the 'Materials and methods'. (D) Table of proteins for the Venn diagram in *Figure 2—figure supplement 1*. Genes encoding the bait proteins are shown in the first three rows. The table contains an alphabetical list ordered by enrichment group (enriched in all bait conditions, enriched in two bait conditions, enriched in one bait condition). To be included in the Venn diagram, enriched is defined as an average ≥1.5-fold number of peptides for a specific protein over the peptide count for the same protein in the BirA-tdTomato condition. (E) Table of proteins for the Venn diagram in *Figure 2—figure supplement 2*. Genes encoding the bait proteins are shown in the first three rows. The table contains an alphabetical list ordered by enrichment group (enriched in all bait conditions, enriched in two bait conditions, enriched in one bait condition). To be included in the Venn diagram, enriched is defined as an average ≥2.5-fold number of peptides for a specific protein over the peptide count for the same protein in the BirA-tdTomato condition. (F) Table of proteins detected in mass spectrometry in RIM cKO$^{DA}$ mice. An alphabetical list of genes encoding the proteins identified by mass spectrometry and associated peptide counts in the RIM cKO$^{DA}$ datasets is provided. (G) Table of proteins detected in mass spectrometry in Syt-1 cKO$^{DA}$ mice. An alphabetical list of genes encoding the proteins identified by mass spectrometry and associated peptide counts in the Syt-1 cKO$^{DA}$ datasets is provided. (H) Table of proteins in RIM cKO$^{DA}$ mice for the Venn diagram in *Figure 3—figure supplement 1*. Genes encoding the bait proteins are shown in the first three rows. The table contains an alphabetical list ordered by enrichment group (enriched in all bait conditions, enriched in two bait conditions, enriched in one bait condition). To be included in the Venn diagram, enriched is defined as an average ≥2.0-fold number of peptides for a specific protein over the peptide count for the same protein in the BirA-tdTomato condition. (I) Table of proteins in Syt-1 cKO$^{DA}$ mice for the Venn diagram in *Figure 3—figure supplement 2*. Genes encoding the bait proteins are shown in the first three rows. The table contains an alphabetical list ordered by enrichment group (enriched in all bait conditions, enriched in two bait conditions, enriched in one bait condition). To be included in the Venn diagram, enriched is defined as an average ≥2.0-fold number of peptides for a specific protein over the peptide count for the same protein in the BirA-tdTomato condition.

## Data availability

All data generated or analyzed in this study are included in the figures and the source data tables. Source data files are provided for Figs. 1 to 3, Fig. 1 - figure supplement 2, Fig. 2 - figure supplements 1 and 2, and Fig. 3 - figure supplements 1 and 2.

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
