## [Editor Report]

Using a smart proximity labeling approach, the protein composition of dopaminergic neurotransmitter release sites was determined in striatal axons. Using mice in which release sites were disrupted as control, the authors identified not only established components of the secretory machinery but also many new proteins whose function awaits further characterization. The datasets provided are of very high quality and provide an important foundation for studies on the dopaminergic exocytotic machinery.

---

## [Decision Letter]

**Decision letter after peer review:**

Thank you for submitting your article "Protein composition of axonal dopamine release sites in the striatum" for consideration by *eLife*. Your article has been reviewed by 3 peer reviewers, including Reinhard Jahn as Reviewing Editor and Reviewer #1, and the evaluation has been overseen by Gary Westbrook as the Senior Editor. The following individual involved in review of your submission has agreed to reveal their identity: Matthijs Verhage (Reviewer #3).

All reviewers agree that the approaches chosen to identify proteins localized to dopaminergic release sites are excellent and cutting-edge. However, the reviewers also agree that while the data provide a valuable and useful resource for future work, both analysis of the data and their interpretation should be carefully reconsidered and revised accordingly:

1. It is not clear to the reviewers, how robust the findings are with respect to the thresholds used for the enrichment analysis (see comments by Ref. 3). For instance, how sensitive are the major trends towards increasing or decreasing the threshold values by e.g. 0.5 points?

2. The discussion should be reconsidered. As pointed out by Ref. 3, it is partially redundant whereas other relevant issues (e.g. completeness of the release site proteome, "oddities" in the data) are not mentioned. Moreover, as pointed out by Ref. 1, a more critical appraisal of the limitations of the chosen labeling procedure should be provided.

*Reviewer #1 (Recommendations for the authors):*

In this study, the protein composition of exocytotic sites in dopaminergic neurons is investigated. While extensive data are available for both glutamatergic and GABA-ergic synapses, it is far less clear which of the known proteins (particularly proteins localized to the active zone) are also required for dopamine release, and whether proteins are involved that are not found in "classical" synapses. The approach used here uses proximity ligation to tag proteins close to synaptic release sites by using three presynaptic proteins (ELKS, RIM, and the beta4-subunit of the voltage-gated calcium channel) as "baits". Fusion proteins containing BirA were selectively expressed in striatal dopaminergic neurons, followed by in-vivo biotin labelling, isolation of biotinylated proteins and proteomics, using proteins labelled after expression of a soluble BirA-construct in dopaminergic neurons as reference. As controls, the same experiments were performed in KO-mouse lines in which the presynaptic scaffolding protein RIM or the calcium sensor synaptotagmin 1 were selectively deleted in dopaminergic neurons. To control for specificity, the proteomes were compared with those obtained by expressing a soluble BirA construct. The authors found selective enrichments of synaptic and other proteins that were disrupted in RIM but not Syt1 KO animals, with some overlap between the different baits, thus providing a novel and useful dataset to better understand the composition of dopaminergic release sites.

Technically, the work is clearly state-of-the-art, cutting-edge, and of high quality, and I have no suggestions for experimental improvements. On the other hand, the data also show the limitations of the approach, and I suggest that the authors discuss these limitations in more detail. The problem is that there is very likely to be a lot of non-specific noise (for multiple reasons) and thus the enriched proteins certainly represent candidates for the interactome in the presynaptic network, but without further corroboration it cannot be claimed that as a whole they all belong to the proteome of the release site.

For instance, in Figure 2 it is shown that a surprisingly high number of postsynaptic proteins are labelled in all three datasets, showing that labelling is not as specific as hoped for (I acknowledge that the authors try to explain this with the presence of neighbouring postsynaptic sites but are they really this close?). The same applies to other proteins that are not expected to be enriched. One of the reasons may be the rather long labelling periods (7 days) which are likely to increase the number of labelled proteins. Recent work from the Rizzoli lab has shown that the half-life of synaptic proteins is around 7-10 days, which means that between 30 and 40 % of the proteins used as baits are turned over during the labelling period. Thus, it is to be expected that labelling is not restricted to the neighbours in the active zone but includes proteins interacting with the baits during their life cycle after biosynthesis and folding at the ribosome, during axonal transport and also during removal from the active zone and transport towards degradative pathways. In this respect, the comparison to the RIM-KO dataset becomes important (Figure 4). In this experiment: what happens to the postsynaptic proteins, to proteins of the presynaptic plasma membrane? Are they still present in the enriched samples?

*Reviewer #2 (Recommendations for the authors):*

The Kaiser lab has been on the forefront in understanding the mechanism of dopamine release in central mammalian neurons. assessing dopamine neuron function has been quite difficult due to the limited experimental access to these neurons. Dopamine neurons possess a number of unique functional roles and participate in several pathophysiological conditions, making them an important target of basic research. This study here has been designed to describe the proteome of the dopamine release apparatus using proximity biotin labeling via active zone protein domains fused to BirA, to test in which ways its proteome composition is similar or different to other central nerve terminals. The control experiments demonstrating proper localization as well as specificity of biotinylation are very solid, yielding in a highly enriched and well characterized proteome data base. Several new proteins were identified and the data base will very likely be a very useful resource for future analysis of the protein composition of synapse and their function at dopamine and other synapses.

The authors find that loss of RIM leads to major reduction in the number of synaptically enriched proteins, while they did not see this loss of number of enriched proteins in the Syt1-KO's, arguing for undisrupted synaptome. Maybe I missed this, but which fraction of proteins and synaptic proteins are than co-detected both in the Syt1 and control conditions when comparing the Venn diagrams of Figure 2 and Figure 3 Suppl. 2? This analysis may provide an estimate of the reliability of the method across experimental conditions.

*Reviewer #3 (Recommendations for the authors):*

In this study Kershberg et al. use three novel in vivo biotin-identification (iBioID) approaches in mice to isolate and identify proteins of axonal dopamine release sites. By dissecting the striatum, where dopamine axons are, from the substantia nigra and VTA, where dopamine somata are, the authors selectively analyzed axonal compartments. Perturbation studies were designed by crossing the iBioID lines with null mutant mice. Combining the data from these three independent iBioID approaches and the fact that axonal compartments are separated from somata provides a precise and valuable description of the protein composition of these release sites, with many new proteins not previously associated with synaptic release sites. These data are a valuable resource for future experiments on dopamine release mechanisms in the CNS and the organization of the release sites. The BirA (BioID) tags are carefully positioned in three target proteins not to affect their localization/function. Data analysis and visualization are excellent. Combining the new iBioID approaches with existing null mutant mice produces powerful perturbation experiments that lead and strong conclusions on the central role of RIM1 as central organizers of dopamine release sites and unexpected (and unexplained) new findings on how RIM1 and synaptotagmin1 are both required for the accumulation of α-synuclein at dopamine release sites.

It is not entirely clear how certain decisions made by the authors on data thresholds may affect the overall picture emerging from their analyses. This is a purely hypothesis-generating study. The authors made little efforts to define expectations and compare their results to these. Consequently, there is little guidance on how to interpret the data and how decisions made by the authors affect the overall conclusions. For instance, the collection of proteins tagged by all three tagging strategies (Figure 2) is expected to contain all known components of dopamine release sites (not at all the case), and maybe also synaptic vesicles (2 TM components detected, but not the most well-known components like vSNAREs and H^+^/DA-transporters), and endocytic machinery (only 2 endophilin orthologs detected). Whether or not a more complete collection the components of release sites, synaptic vesicles or endocytic machinery are observed might depend on two hard thresholds applied in this study: (a) "Hits" (depicted in Figure 2) were defined as proteins enriched {greater than or equal to} 2-fold (line 178) and peptides not detected in the negative control (soluble BirA) were defined as 0.5 (line 175). How crucial are these two decisions? It would be great to know if the overall conclusions change if these decisions were made differently.

Given the good separation of the axonal compartment from the somata (one of the real experimental strengths of this study), it is completely unexpected to find two histones being enriched with all three tagging strategies (Hist1h1d and 1h4a). This should be mentioned and discussed.

It would also help to compare the data more systematically to a previous study that attempted to define release sites (albeit not dopamine release sites) using a different methodology (biochemical purification): Boyken et al. (only mentioned in relation to Nptn, but other proteins are observed in both studies too, e.g. Cend1).

The Discussion section could be improved. It is rather long (7.5p) with some repetition from the Introduction, some issues already demonstrated convincingly before (2.5p on RIM being a scaffold in DA neurons) and a rather speculative discussion on RIM being associated with Parkinson's disease through dopamine release site recruitment of α-synuclein, possibly via protein interactions or through vesicle docking (but how is release site recruitment of α-synuclein important for protein interactions, vesicle docking or Parkinson symptoms?). Instead some of the more pressing questions (at least to this reviewer) are not discussed, such as the expectations mentioned above: why were expected components of release sites not detected, why only some but not all expected components of synaptic vesicles/endocytic machinery and why are two histones in the axonal compartment?

---

## [Author Response]

1. It is not clear to the reviewers, how robust the findings are with respect to the thresholds used for the enrichment analysis (see comments by Ref. 3). For instance, how sensitive are the major trends towards increasing or decreasing the threshold values by e.g. 0.5 points?

We agree that thresholding criteria are critical decisions, both for our work and for papers using similar techniques. To illustrate why we chose 2.0 as an enrichment threshold beyond justification through literature ^1,2^, we have included new figures showing analyses of the main release site proteome at varying thresholds. Figure 2 presents the 2.0 enrichment threshold, while Figure 2 —figure supplements 1 and 2 present the same data with 1.5 and 2.5 thresholds, respectively. Naturally, the overall size of the release site proteome varies considerably with 527 hits at 2.0, 991 hits at 1.5, and 348 hits at 2.5 enrichment thresholds. We then took each thresholded proteome and analyzed it with SynGO to determine the fraction of proteins previously associated with synapses (Figure 2 —figure supplement 3). SynGO-annotated proteins were 37% (194 of 527) for 2.0, 31% (304 of 991) for 1.5, and 33% (116 of 348) for 2.5. Based on these analyses, and consistent with literature on other iBioID experiments, we think it is appropriate to select 2.0 as the threshold for further analyses and comparisons. The data that led to this decision are now fully incorporated in the manuscript, including the figures (Figure 2 —figure supplements 1 to 3) and a new paragraph in the Results section on lines 208-215. We further have added a summary of limitations together with conclusions in the last section of the discussion that includes the point of thresholding. This section is captioned “Summary of conclusions and limitations” (lines 501-518).

To complement the presentation of the control proteome with various thresholds, we also determined the number of hits in the release site proteomes of RIM cKO^DA^ and Syt-1 cKO^DA^ mice at these thresholds. The major finding that RIM ablation disrupts this proteome was independent of thresholding, with 268 hits at 1.5, 198 at 2.0 and 82 hits at 2.5 enrichment thresholds. The overall number of hits in Syt-1 cKO^DA^ mice remained higher, with 602, 450 and 252 at the 1.5, 2.0 and 2.5 enrichment thresholds, respectively. This is included in the revised manuscript on lines 298-301. Overall, the main findings are not affected by thresholding, but the numbers for proteins in each proteome vary based on threshold.

2. The discussion should be reconsidered. As pointed out by Ref. 3, it is partially redundant whereas other relevant issues (e.g. completeness of the release site proteome, "oddities" in the data) are not mentioned. Moreover, as pointed out by Ref. 1, a more critical appraisal of the limitations of the chosen labeling procedure should be provided.

We agree, and we have made the requested adjustments. Overall, we have shortened the discussion from 2400 words to 1900 words. We have removed redundancies between the discussion and the Results section and have included additional discussion of unexpected hits (for example: histones, lines 482-485; transport proteins, lines 493-497). We have also included a section on summarizing limitations, together with key conclusions, at the end of the discussion (lines 501-518). We specifically include a more in-depth discussion of noise, of oddities, of thresholding and of the fundamental point that the proteome is based on proximity, not on interaction.

Reviewer #1 (Recommendations for the authors):In this study, the protein composition of exocytotic sites in dopaminergic neurons is investigated. While extensive data are available for both glutamatergic and GABA-ergic synapses, it is far less clear which of the known proteins (particularly proteins localized to the active zone) are also required for dopamine release, and whether proteins are involved that are not found in "classical" synapses. The approach used here uses proximity ligation to tag proteins close to synaptic release sites by using three presynaptic proteins (ELKS, RIM, and the beta4-subunit of the voltage-gated calcium channel) as "baits". Fusion proteins containing BirA were selectively expressed in striatal dopaminergic neurons, followed by in-vivo biotin labelling, isolation of biotinylated proteins and proteomics, using proteins labelled after expression of a soluble BirA-construct in dopaminergic neurons as reference. As controls, the same experiments were performed in KO-mouse lines in which the presynaptic scaffolding protein RIM or the calcium sensor synaptotagmin 1 were selectively deleted in dopaminergic neurons. To control for specificity, the proteomes were compared with those obtained by expressing a soluble BirA construct. The authors found selective enrichments of synaptic and other proteins that were disrupted in RIM but not Syt1 KO animals, with some overlap between the different baits, thus providing a novel and useful dataset to better understand the composition of dopaminergic release sites.Technically, the work is clearly state-of-the-art, cutting-edge, and of high quality, and I have no suggestions for experimental improvements.

We thank the reviewer for this summary and for pointing out the high quality of the work.

On the other hand, the data also show the limitations of the approach, and I suggest that the authors discuss these limitations in more detail. The problem is that there is very likely to be a lot of non-specific noise (for multiple reasons) and thus the enriched proteins certainly represent candidates for the interactome in the presynaptic network, but without further corroboration it cannot be claimed that as a whole they all belong to the proteome of the release site.

We fully agree with the reviewer. Most importantly, we have changed the final section from “Conclusions” to “Summary of conclusions and limitations” (lines 501-518) to summarize the limitations with equal weight to the conclusions. In the revised manuscript, we also included many specific additional points in this respect throughout the discussion and the results: many hits could be noise (lines 458, 478-479), thresholding affects the inclusion of proteins in the release site dataset (lines 208-215), the seven-day time window could deliver interactors from the soma to the synapse (lines 493-495), specific oddities (for example histones, lines 482-485), iBioID does not deliver an interactome *per se* but is simply based on proximity (lines 505-508), and several more. We also clearly state that each specific hit needs follow-up studies (lines 501-503: *“Each protein will require validation through morphological and functional characterization before an unequivocal assignment to dopamine release sites is possible.*”), and a similar statement was added on lines 374-375.

For instance, in Figure 2 it is shown that a surprisingly high number of postsynaptic proteins are labelled in all three datasets, showing that labelling is not as specific as hoped for (I acknowledge that the authors try to explain this with the presence of neighbouring postsynaptic sites but are they really this close?). The same applies to other proteins that are not expected to be enriched. One of the reasons may be the rather long labelling periods (7 days) which are likely to increase the number of labelled proteins. Recent work from the Rizzoli lab has shown that the half-life of synaptic proteins is around 7-10 days, which means that between 30 and 40 % of the proteins used as baits are turned over during the labelling period. Thus, it is to be expected that labelling is not restricted to the neighbours in the active zone but includes proteins interacting with the baits during their life cycle after biosynthesis and folding at the ribosome, during axonal transport and also during removal from the active zone and transport towards degradative pathways. In this respect, the comparison to the RIM-KO dataset becomes important (Figure 4). In this experiment: what happens to the postsynaptic proteins, to proteins of the presynaptic plasma membrane? Are they still present in the enriched samples?

We fully agree with the reviewer and have added these points to the results, discussion, and limitations sections mentioned above.

The RIM cKO^DA^ and control datasets reveal that axonal proteins assessed with BirA-tdTomato have high overlap, with 620 out of the 671 found in control also present in RIM cKO^DA^. This suggests that the overall protein content is very similar. Importantly, in the release site dataset, presynaptic proteins decrease from 99 hits in the control to 43 hits in RIM cKO^DA^. Synaptic vesicle proteins decreased from 30 to 11, active zone proteins from 14 to 4, and postsynaptic proteins from 87 to 37. We now added these numbers to the legend of Figure 3 —figure supplement 1 and discuss postsynaptic proteins on lines 457-475. We have now also added analyses of postsynaptic proteins identical to how we assessed synaptic, presynaptic and active zone proteins. As we observed for the presynaptic and active zone annotations, RIM cKO^DA^ depleted postsynaptic proteins. This supports that enrichment of these proteins is specific and does not simply reflect noise. These new analyses are now included in Figure 4 —figure supplement 1.

These analyses support that the hits in the control dataset are generally specific and that the postsynaptic hits are not just noise; if they were noise, they should be similar between the control proteome and the RIM cKO^DA^ proteome. These proteins could either be detected because they are also present in dopamine axons, and/or because iBioID biotinylates proteins across membranes as long as they are close enough. We suspect that both factors contribute. We think that the latter point also matches with data from others: if PSD proteins are used as iBioID baits, synaptic vesicle and active zone proteins are generally detected ^2^. Ultimately, we think that this may reflect that the pre- and postsynaptic compartments are within the labeling radius of each other and that reactive biotin can pass through lipid bilayers. The latter point is also vividly demonstrated by the notion that our and published proteomes contain intra-mitochondrial proteins ^2,3^.

Reviewer #2 (Recommendations for the authors):The Kaiser lab has been on the forefront in understanding the mechanism of dopamine release in central mammalian neurons. assessing dopamine neuron function has been quite difficult due to the limited experimental access to these neurons. Dopamine neurons possess a number of unique functional roles and participate in several pathophysiological conditions, making them an important target of basic research. This study here has been designed to describe the proteome of the dopamine release apparatus using proximity biotin labeling via active zone protein domains fused to BirA, to test in which ways its proteome composition is similar or different to other central nerve terminals. The control experiments demonstrating proper localization as well as specificity of biotinylation are very solid, yielding in a highly enriched and well characterized proteome data base. Several new proteins were identified and the data base will very likely be a very useful resource for future analysis of the protein composition of synapse and their function at dopamine and other synapses.

We thank the reviewer for this positive assessment of our work.

The authors find that loss of RIM leads to major reduction in the number of synaptically enriched proteins, while they did not see this loss of number of enriched proteins in the Syt1-KO's, arguing for undisrupted synaptome. Maybe I missed this, but which fraction of proteins and synaptic proteins are than co-detected both in the Syt1 and control conditions when comparing the Venn diagrams of Figure 2 and Figure 3 Suppl. 2? This analysis may provide an estimate of the reliability of the method across experimental conditions.

We thank the reviewer for proposing to be clear in the comparison of the control and Syt-1 cKO^DA^ data. A direct comparison of hit numbers is included on lines 323-324, with 37% overlap between control and Syt-1 cKO^DA^ (vs. 15% between control and RIM cKO^DA^). A direct mapping of this overlap is included in Figure 4E. We think that this direct comparison is complicated by a number of factors, as outlined below, and did our best to include these complications in the discussion, including the last section (lines 501-518).

First, to assess overall similarity, the initial comparison should be to assess axonal proteins identified in the BirA-tdTomato samples. These datasets are quite similar, with 671 (control) and 793 (Syt-1 cKO^DA^) proteins detected, and a high overlap of 601 proteins. We think that this indicates that the experiment *per se* is quite reproducible. The comparison of the release site proteome between control and Syt-1 cKO^DA^ is more complicated. We think that the main point of this comparison is that the overall number of hits is quite similar, with 450 hits in the Syt-1 cKO^DA^ proteome and 527 hits in the control proteome, and we now show that this similarity holds across multiple thresholds (lines 298-301; ≥ 1.5: Syt-1 cKO^DA^ 602 hits, control 991, ≥ 2.0: 450/527, ≥ 2.5: 252/348). Detailed analyses of overlap reveals that known active zone proteins such as Bassoon, Ca_V_2 channels, RIMs, and ELKS proteins are present in both proteomes, but the overlap is partial and incomplete with 191 proteins found in both proteomes. As discussed throughout and summarized on lines 501-518, the reasons for this partial overlap may be manifold. Trivially, it could be explained by noise or non-saturation (“*incompleteness*”) of the proteome. We also think that the Syt-1 proteome is not expected to be identical because there is a strong release deficit in these mice. If Syt-1 has a dopamine vesicle docking function (which it does at conventional synapses ^4^), this could influence the proteome. We note that protein functions in the dopamine axon are not well established, but inferred from studies of classical synapses.

We have scrutinized the manuscript to not express that the control and Syt-1 cKO^DA^ proteomes are identical; we know they are not and discuss the example of α-synuclein specifically (Figure 6, lines 347-362). Rather, the striking part is that the extent of the proteomes with high hit number, much higher than RIM cKO^DA^, are similar. Specific hits have to be assessed in a detailed way, one hit at a time, in future studies, as expressed unequivocally on lines 501-503).

Reviewer #3 (Recommendations for the authors):In this study Kershberg et al. use three novel in vivo biotin-identification (iBioID) approaches in mice to isolate and identify proteins of axonal dopamine release sites. By dissecting the striatum, where dopamine axons are, from the substantia nigra and VTA, where dopamine somata are, the authors selectively analyzed axonal compartments. Perturbation studies were designed by crossing the iBioID lines with null mutant mice. Combining the data from these three independent iBioID approaches and the fact that axonal compartments are separated from somata provides a precise and valuable description of the protein composition of these release sites, with many new proteins not previously associated with synaptic release sites. These data are a valuable resource for future experiments on dopamine release mechanisms in the CNS and the organization of the release sites. The BirA (BioID) tags are carefully positioned in three target proteins not to affect their localization/function. Data analysis and visualization are excellent. Combining the new iBioID approaches with existing null mutant mice produces powerful perturbation experiments that lead and strong conclusions on the central role of RIM1 as central organizers of dopamine release sites and unexpected (and unexplained) new findings on how RIM1 and synaptotagmin1 are both required for the accumulation of α-synuclein at dopamine release sites.

We thank the reviewer for assessing our paper, for summarizing our main findings, and for expressing genuine enthusiasm for the approach and the outcomes.

It is not entirely clear how certain decisions made by the authors on data thresholds may affect the overall picture emerging from their analyses. This is a purely hypothesis-generating study. The authors made little efforts to define expectations and compare their results to these. Consequently, there is little guidance on how to interpret the data and how decisions made by the authors affect the overall conclusions. For instance, the collection of proteins tagged by all three tagging strategies (Figure 2) is expected to contain all known components of dopamine release sites (not at all the case), and maybe also synaptic vesicles (2 TM components detected, but not the most well-known components like vSNAREs and H^+^/DA-transporters), and endocytic machinery (only 2 endophilin orthologs detected). Whether or not a more complete collection the components of release sites, synaptic vesicles or endocytic machinery are observed might depend on two hard thresholds applied in this study: (a) "Hits" (depicted in Figure 2) were defined as proteins enriched {greater than or equal to} 2-fold (line 178) and peptides not detected in the negative control (soluble BirA) were defined as 0.5 (line 175). How crucial are these two decisions? It would be great to know if the overall conclusions change if these decisions were made differently.

We agree with the reviewer that the thresholding decisions are important and have now better incorporated the rationale for these decisions in the manuscript.

Two-fold enrichment threshold. As outlined in the response to point 1 in the editorial decision letter, we now include figure supplements to illustrate the composition of the control proteome if we apply 1.5- or 2.5-fold enrichment thresholds (Figure 2 —figure supplements 1 and 2) instead of the 2.0-fold threshold used in Figure 2. This leads to more or less hits (991 and 348, respectively) compared to the 2.0-fold threshold (527 hits). It is noteworthy that the SynGO-overlap is the highest with the 2.0 threshold (37% vs. 31% at 1.5 and 33% at 2.5, Figure 2 —figure supplement 3), justifying this threshold experimentally in addition to what was done in previous work ^1,2^. These data are now described on lines 208-215 of the manuscript. When we apply these different thresholds to RIM and Syt-1 cKO^DA^ datasets, the finding that RIM ablation disrupts release site assembly persists. The following hit numbers were observed in the mutants at the 1.5, 2.0 and 2.5 enrichment thresholds, respectively: RIM cKO^DA^ 268, 198 and 82 hits; Syt cKO^DA^ 602, 450 and 252 hits. Hence, the extent of the release site proteome remains much smaller after RIM ablation independent of the enrichment threshold, bolstering the conclusion that RIM is an important scaffold for these release sites. This is included in the revised manuscript on lines 298-301.

Undetected peptides in BirA-tdTomato. We did not express this well enough in the manuscript. The undetected proteins were set to 0.5 such that a protein that was detected with a specific bait but not with BirA-tdTomato could be illustrated with a specific circle size, not to determine inclusion in the analyses. If the average peptide count across repeats with a specific bait was 1, this resulted in inclusion in Figure 2 and consecutive analyses with the smallest circle size. Hence, this decision was made to define circle size. It did not affect inclusion in Figure 2 beyond the following two points. If one were to further decrease it, this might result in including peptides that only appeared once as a single peptide for some of the experiments, which we wanted to avoid. If one would set it higher (to 1), this artificial threshold would be equal to proteins that were actually detected experimentally multiple times, which we wanted to avoid as well. We have now clarified this on lines 165-167 and lines 1119-1121.

Expected proteins. In general, interpreting our dataset with a strong prior of expected proteins is difficult. The literature on release site proteins specifically characterized for dopamine is limited. We have found Bassoon, RIM, ELKS and Munc13 to be present using 3D-SIM super-resolution microscopy ^5,6^, and we indeed found these proteins in the data as discussed on lines 227-232 and lines 423-445 in the revised manuscript. The prediction for vesicular and endocytic proteins is complicated. Release sites are sparse ^5,7^, and vesicle clusters are widespread in the dopamine axon, in some cases filling most of the axon (for example, see extended vesicle clusters filling much of the dopamine axon in Figure 7E of ^5^). Furthermore, docking in dopamine axons has not been characterized, and it is unclear how frequently vesicles are docked. Hence, it is not clear whether vesicular proteins should be concentrated at release sites compared to the rest of the axon (the BirA-tdTomato proteome we use for normalization). Similar points can be made for proteins for endocytosis and recycling of dopamine vesicles. Within the dopamine system, it is unclear whether the recycling pathway is close to the exocytic sites. One consistent finding across functional studies is that depletion after activity is unusually long-lasting in the dopamine system, for tens of seconds, even after only mild stimulation ^5,8–13^. Hence, endocytosis and RRP replenishment might be very slow in these axons. It is not certain that endocytic factors are predeployed to the plasma membrane, and if they are, it is unclear how close to release sites they would be. As such, we agree with the reviewer that the proteome we describe is a hypothesis-generator. With the limited knowledge on dopamine release, predictions beyond the previously characterized proteins in dopamine axons are difficult to make.

We thank the reviewer for suggesting to include a better analysis of different thresholds and for giving us the opportunity to clarify the other points that were raised.

Given the good separation of the axonal compartment from the somata (one of the real experimental strengths of this study), it is completely unexpected to find two histones being enriched with all three tagging strategies (Hist1h1d and 1h4a). This should be mentioned and discussed.

We agree with the reviewer and have addressed this point in the manuscript. This could either reflect noise, or there could be more specific reasons behind it. The manuscript now states on lines 482-485: “*It is surprising that Hist1h1d and Hist1h4a, genes encoding for the histone proteins H1.3 and H4, were robustly enriched (Figure 2A). These hits might be entirely unspecific, or their co-purification could be due to biotinylation of H1 and H4 proteins (Stanley et al., 2001). It is also possible that there are unidentified synaptic functions of some of the unexpected proteins.*” Ultimately, we do not know why these proteins are enriched, and we state clearly in the section “Summary of conclusions and limitations” that each new hit has to be validated in future studies (lines 501-503).

It would also help to compare the data more systematically to a previous study that attempted to define release sites (albeit not dopamine release sites) using a different methodology (biochemical purification): Boyken et al. (only mentioned in relation to Nptn, but other proteins are observed in both studies too, e.g. Cend1).

We agree with the reviewer that Boyken et al., 2013 ^14^ is an important resource for our paper and for the assessment of the proteomic composition of release sites. We have now introduced links and citations to this paper multiple times (for example, on lines 231, 241, 430, 443, 481) and have expanded the discussion of overlap between these proteomes, including on Cend1 (lines 479482).

We think that a systematic comparison with Boyken et al., 2013 ^14^ is complicated because (1) so little is known about dopamine release mechanics and (2) because the approach is very different between the two papers. In respect to (1), most prominently, it is not certain how frequently vesicles are docked in the dopamine axon. Only ~25% of the varicosities contain these release sites, and vesicle docking has not been characterized in striatal dopamine axons to the best of our knowledge. Hence, how a docking site at a classical synapse compares to a dopamine release site remains unclear at the outset. For point (2), the key difference is that “within dataset normalizations” are very different in these two studies. In our iBioID dataset, we normalize to soluble proteins defined as proximity to BirA-tdTomato. In ref. ^14^, the authors express enrichment over “light”, regular synaptic vesicles purified with the same approach. This has a major impact on the proteome that strongly influences a direct comparison of hits, because there are large differences in the normalization. While each normalization makes sense for the respective paper, it complicates direct comparison.

With these points in mind, we have compared hits across both datasets class-by-class. For some classes, the datasets have reasonable overlap for ≥ 2-fold enriched proteins: for example for active zone proteins (3 of 7 hits in ^14^ appear in our control proteome) and adhesion and cell surface proteins (8 of 18). For other classes, the overlap is limited: for example for nucleotide metabolism/protein synthesis (0 of 16 hits in ^14^ appear in our dataset) and cytoskeletal proteins (5 of 29). We hope the reviewer agrees, that given these factors, the analyses and discussion needed for a systematic comparison goes beyond the scope of our paper. We have instead added a number of references to Boyken et al., 2013 ^14^, as outlined above, when direct comparison is meaningful.

The Discussion section could be improved. It is rather long (7.5p) with some repetition from the Introduction, some issues already demonstrated convincingly before (2.5p on RIM being a scaffold in DA neurons) and a rather speculative discussion on RIM being associated with Parkinson's disease through dopamine release site recruitment of α-synuclein, possibly via protein interactions or through vesicle docking (but how is release site recruitment of α-synuclein important for protein interactions, vesicle docking or Parkinson symptoms?). Instead some of the more pressing questions (at least to this reviewer) are not discussed, such as the expectations mentioned above: why were expected components of release sites not detected, why only some but not all expected components of synaptic vesicles/endocytic machinery and why are two histones in the axonal compartment?

We have re-written the discussion. Specifically, we have (1) shortened it from ~2400 to ~1900 words, (2) cut down the RIM scaffolding section from 2.5 pages to 1 page, (3) removed the repeated discussion of α-synuclein as it is redundant with the Results section, (4) added more discussion on unexpected components (for example histones on lines 482-485, trafficking components on lines 493-497, and ribosomes on lines 497-498), (5) expanded the section on predicted active zone components (lines 423-445), and (6) more clearly described the limitations of the method throughout the manuscript (for example on lines 458, 478-479 for noise, on lines 208-215 for thresholding) and summarize this point in the last section (lines 501-518).

It is difficult to make specific predictions on vesicular and endocytic proteins, and we think those components are not necessarily expected to be enriched over their general axonal levels within 50 nm of the baits at rest. Vesicular proteins may be high within the rest of the axon because the axon is densely packed with vesicles and release sites are sparse ^5,7,15^, and it is unclear how frequent dopamine vesicle docking is. It is also unclear whether endocytic machinery is predeployed to sites of exocytosis. Replenishment is astonishingly slow in dopamine axons (tens of seconds to a minute) ^5,8–13^, it is uncertain where endocytosis happens relative to release sites, and it is unclear whether endocytic factors are concentrated at the release site compared to their average axonal protein content (detected with BirA-tdTomato). Hence, we think it is better to not describe the proteome with a prior, but to interpret it as it is. We hope that the reviewer agrees in light of these explanations and with the numerous adjustments that we have made to the discussion. There are strong predictions based on previous dopamine release site analyses what active zone proteins should be found at these sites, and we discuss them in detail (lines 423-445). Beyond these proteins, predictions are difficult to make.

References

1. Takano, T.; Wallace, J. T.; Baldwin, K. T.; Purkey, A. M.; Uezu, A.; Courtland, J. L.; Soderblom, E. J.; Shimogori, T.; Maness, P. F.; Eroglu, C.; Soderling, S. H. ChemicoGenetic Discovery of Astrocytic Control of Inhibition in vivo. Nature 2020, 588, 296–302.

2. Uezu, A.; Kanak, D. J.; Bradshaw, T. W. A.; Soderblom, E. J.; Catavero, C. M.; Burette, A. C.; Weinberg, R. J.; Soderling, S. H. Identification of an Elaborate Complex Mediating Postsynaptic Inhibition. Science 2016, 353, 1123–1129.

3. Loh, K. H.; Stawski, P. S.; Draycott, A. S.; Udeshi, N. D.; Lehrman, E. K.; Wilton, D. K.; Svinkina, T.; Deerinck, T. J.; Ellisman, M. H.; Stevens, B.; Carr, S. A.; Ting, A. Y. Proteomic Analysis of Unbounded Cellular Compartments: Synaptic Clefts. Cell 2016, 166, 12951307.e21.

4. Chang, S.; Trimbuch, T.; Rosenmund, C. Synaptotagmin-1 Drives Synchronous ca^2+^Triggered Fusion by C2B-Domain-Mediated Synaptic-Vesicle-Membrane Attachment. Nature Neuroscience 2018, 21, 33–40.

5. Liu, C.; Kershberg, L.; Wang, J.; Schneeberger, S.; Kaeser, P. S. Dopamine Secretion Is Mediated by Sparse Active Zone-like Release Sites. Cell 2018, 172, 706-718.e15.

6. Banerjee, A.; Imig, C.; Balakrishnan, K.; Kershberg, L.; Lipstein, N.; Uronen, R.-L.; Wang, J.; Cai, X.; Benseler, F.; Rhee, J. S.; Cooper, B. H.; Liu, C.; Wojcik, S. M.; Brose, N.; Kaeser, P. S. Molecular and Functional Architecture of Striatal Dopamine Release Sites. Neuron 2022, 110, 248-265.e9.

7. Pereira, D. B.; Schmitz, Y.; Mészáros, J.; Merchant, P.; Hu, G.; Li, S.; Henke, A.; LizardiOrtiz, J. E.; Karpowicz, R. J.; Morgenstern, T. J.; Sonders, M. S.; Kanter, E.; Rodriguez, P. C.; Mosharov, E. V; Sames, D.; Sulzer, D. Fluorescent False Neurotransmitter Reveals Functionally Silent Dopamine Vesicle Clusters in the Striatum. Nature Neuroscience 2016, 19, 578–586.

8. Banerjee, A.; Lee, J.; Nemcova, P.; Liu, C.; Kaeser, P. S. Synaptotagmin-1 Is the ca^2+^ Sensor for Fast Striatal Dopamine Release. *eLife* 2020, 9.

9. Wang, L.; Shang, S.; Kang, X.; Teng, S.; Zhu, F.; Liu, B.; Wu, Q.; Li, M.; Liu, W.; Xu, H.; Zhou, L.; Jiao, R.; Dou, H.; Zuo, P.; Zhang, X.; Zheng, L.; Wang, S.; Wang, C.; Zhou, Z. Modulation of Dopamine Release in the Striatum by Physiologically Relevant Levels of Nicotine. Nature communications 2014, 5, 3925.

10. Brimblecombe, K. R.; Gracie, C. J.; Platt, N. J.; Cragg, S. J. Gating of Dopamine Transmission by Calcium and Axonal N-, Q-, T- and L-Type Voltage-Gated Calcium Channels Differs between Striatal Domains. The Journal of Physiology 2015, 593, 929– 946.

11. Marcott, P. F.; Mamaligas, A. A.; Ford, C. P. Phasic Dopamine Release Drives Rapid Activation of Striatal D2-Receptors. Neuron 2014, 84, 164–176.

12. Patriarchi, T.; Cho, J. R.; Merten, K.; Howe, M. W.; Marley, A.; Xiong, W.-H.; Folk, R. W.; Broussard, G. J.; Liang, R.; Jang, M. J.; Zhong, H.; Dombeck, D.; Zastrow, M. von; Nimmerjahn, A.; Gradinaru, V.; Williams, J. T.; Tian, L. Ultrafast Neuronal Imaging of Dopamine Dynamics with Designed Genetically Encoded Sensors. Science 2018, 360, eaat4422.

13. Wang, L.; Zhang, X.; Xu, H.; Zhou, L.; Jiao, R.; Liu, W.; Zhu, F.; Kang, X.; Liu, B.; Teng, S.; Wu, Q.; Li, M.; Dou, H.; Zuo, P.; Wang, C.; Wang, S.; Zhou, Z. Temporal Components of Cholinergic Terminal to Dopaminergic Terminal Transmission in Dorsal Striatum Slices of Mice. The Journal of Physiology 2014, 592, 3559–3576.

14. Boyken, J.; Grønborg, M.; Riedel, D.; Urlaub, H.; Jahn, R.; Chua, J. J. Molecular Profiling of Synaptic Vesicle Docking Sites Reveals Novel Proteins but Few Differences between Glutamatergic and GABAergic Synapses. Neuron 2013, 78, 285–297.

15. Wildenberg, G.; Sorokina, A.; Koranda, J.; Monical, A.; Heer, C.; Sheffield, M.; Zhuang, X.; McGehee, D.; Kasthuri, B. Partial Connectomes of Labeled Dopaminergic Circuits Reveal Non-Synaptic Communication and Axonal Remodeling after Exposure to Cocaine. *eLife* 2021, 10.